# Dynamics of the Askja caldera July 2014 landslide, Iceland, from seismic signal analysis: precursor, motion and aftermath

Anne Schöpa[1], Wei-An Chao[2], Bradley P. Lipovsky[3], Niels Hovius[1,4], Robert S. White[5], Robert G. Green[5,1], Jens M. Turowski[1]

[1]Helmholtz Centre Potsdam GFZ German Research Centre for Geosciences, 14473 Potsdam, Germany
[2]Department of Civil Engineering, National Chiao Tung University, Hsinchu 30010, Taiwan
[3]Department of Earth and Planetary Sciences, Harvard University, Cambridge, MA 02138, USA
[4]Institute of Earth and Environmental Science, University of Potsdam, 14476 Potsdam, Germany
[5]Department of Earth Sciences. University of Cambridge, Cambridge CB3 0EZ, UK

*Correspondence to*: Anne Schöpa (schoepa@gfz-potsdam.de)

**Abstract.** Landslide hazard motivates the need for a deeper understanding of the events that occur before, during and after catastrophic slope failures. Due to the destructive nature of such events, in situ observation is often difficult or impossible. Here, we use data from a network of 58 seismic stations to characterise a large landslide at the Askja caldera, Iceland, on 21 July 2014. High data quality and extensive network coverage allow us to analyse both long- and short-period signals associated with the landslide, and thereby obtain information about its triggering, initiation, timing and propagation. At long periods, a landslide force history inversion shows that the Askja landslide was a single, large event starting at the SE corner of the caldera lake at 23:24:05 UTC and propagating to the NW in the following 2 minutes. The bulk sliding mass was 7–16×10$^{10}$ kg, equivalent to a collapsed volume of 35–80×10$^6$ m$^3$. The sliding mass was displaced downslope by 1260 ± 250 m. At short periods, seismic tremor was observed for 30 minutes before the landslide. The tremor is approximately harmonic with a fundamental frequency of 2.3 Hz and shows time-dependent changes of its frequency content. We attribute the seismic tremor to stick-slip motion along the landslide failure plane. Accelerating motion leading up to the catastrophic slope failure culminated in an aseismic quiescent period for two minutes before the landslide. We propose that precursory seismic signals may be useful in landslide early-warning systems. The 8 hours after the main landslide failure are characterised by smaller slope failures originating from the destabilised caldera wall decaying in frequency and magnitude. We introduce the term afterslides for this subsequent, declining slope activity after a large landslide.

## 1 Introduction

Volcanic edifices are prone to landsliding because of their usually steep topography, fresh, unconsolidated deposits, and high seismic, volcanic and hydrothermal activity, and the associated surface deformation. In the past, tsunami-creating landslides at volcanic edifices have led to the destruction of infrastructure and high numbers of fatalities. For example, the 1792 Unzen Mayu-Yama, Japan, landslide and the resulting tsunami killed more than 15,000 people in the Shimabara Bay (Sassa et al.,

2016) and the eruption of Mt. St. Helens, USA, on 18 May 1980 initiated a $2.3 \times 10^9$ m$^3$ landslide that ran into Spirit Lake and caused a 260-m-high wave deforesting adjoining slopes (Voight et al., 1981). Seismic networks are often installed around volcanoes for monitoring of magmatic processes and eruption forecasting. Their seismic records can also hold valuable information about landslide events occurring on the edifice.

Seismic signals of landslides are a powerful tool to reconstruct the dynamics of the slope failure including source mechanisms, the failure sequence together with precursory activity, and landslide properties (Brodsky et al., 2003; Favreau et al., 2010; Schneider et al., 2010; Moretti et al., 2012; Allstadt, 2013; Yamada et al., 2013). Long-period seismic signals of landslides from stations several thousand kilometres away can be used as references for inversions (Allstadt, 2013; Ekström and Stark, 2013; Yamada et al., 2013; Hibert et al., 2015; Chao et al., 2016) or models (Brodsky et al., 2003; Favreau et al.,
2010; Schneider et al., 2010; Moretti et al., 2012) to constrain the location, mass, duration, displacement, and run-out trajectories of the landslide. Short-period waves, generated by the momentum exchanges within a granular landslide mass and along its boundaries, have been used to study the detachment, moving and reposing phases of landslides (Norris, 1994; Suriñach et al., 2005; Dammeier et al., 2011; Hibert et al., 2011, 2014; Deparis et al., 2008; Vilajosana et al., 2008). Seismic records can also give valuable information about triggers and precursors of slope failures (Amitrano et al., 2005; Caplan-
Auerbach and Huggel, 2007; Senfaute et al., 2009; Got et al., 2010; Helmstetter and Garambois, 2010; Dietze et al., 2017). For instance, repeated small earthquakes indicative of stick-slip movement on a small patch were observed before a landslide failed within shale and tuff layers in Rausu, Japan (Yamada et al., 2016). Additionally, individual cracking events that occur more frequently in time closer to the main failure were identified at a station 200 m away from the steep, bedrock source area of a $10^4$ m$^3$ landslide in the Illgraben, Switzerland (Zeckra et al., 2015). However, the localisation and characterisation
of precursory slope activity before large mass wasting events is often limited by sparse seismic station coverage, preventing a detailed analysis of the underlying source mechanisms.

In this study, we present seismic data from the 2014 Askja landslide. As the landslide was located in the centre of a temporary local network of 58 seismic stations the spatial coverage is exceptionally good and the signal-to-noise ratio (SNR) of most stations is high due to their remote locations far away from roads or other places of human activity. The high seismic
data quality allows for a detailed reconstruction of the landslide dynamics based on the combined analysis of records from stations within a few kilometres of the landslide and at distances of up to 100 km. A force history inversion of the long-period signals of the landslide from distant seismic stations of the network is used to infer its timing, propagation direction, mass, and vertical and horizontal displacement. The short-period signals of nearby stations are included in a comprehensive interpretation of the landslide dynamics. These signals also contain information about the processes occurring before and
after the catastrophic slope failure. We identified a precursory tremor signal in the seismic data of 38 stations located up to 30 km away from the landslide source area starting 30 minutes before the large landslide. We find the most plausible explanation of the precursory tremor to be repeated stick-slip motion along the landslide failure plane in moments preceding catastrophic failure. After the catastrophic failure of the landslide, the seismic stations a few kilometres from the landslide

source area recorded smaller slope failures. These smaller rockfalls and slides initiated from the destabilised section of the caldera wall where the large landslide originated.

## 2 The Askja landslide and its failure preconditions

In the following, we first report on the Askja landslide and introduce the reader to the Askja area before we describe the factors that made the landslide source area prone to slope failure. We then focus on the seismic dataset and use it to characterise the landslide, the precursory tremor and the subsequent small slope failures.

In the late evening of 21 July 2014, a white cloud was seen rising over the Askja central volcano in the Icelandic highlands (Helgason et al., 2014). As field investigations on the following days showed, the cloud was a consequence of a voluminous landslide, which must have occurred during the night. The cloud probably contained a mixture of dust from the landslide and steam from the hydrothermal system at the south-eastern shore of the Askja caldera lake Öskjuvatn that was depressurised by the removal of the landslide mass (Vogfjörd et al., 2015). At this south-eastern lakeshore, steep scars and fresh, mobilised material could be seen (Hoskuldsson et al., 2015). Parts of the landslide material must have entered the lake and created tsunami waves as flood marks up to 60–80 m above the lake level were found at the shorelines (Gylfadóttir et al., 2017). The flood marks implied that up to ten individual waves inundated the shore and also went into the 200 m wide Víti crater, a popular tourist spot on the north-eastern side of the lake. Analysis of the seismic record of the permanent stations of the Icelandic Meteorological Institute showed that the landslide occurred at 23:24 UTC, equivalent to local time (Saemundsson et al., 2015). This timing meant that no eyewitnesses were present. Geodetic surveys estimated the landslide volume to be $12–50\times10^6$ m$^3$ and that about $10\times10^6$ m$^3$ entered the caldera lake creating the tsunami waves (Gylfadóttir et al., 2017).

Several factors made the site at the south-eastern corner of Lake Öskjuvatn prone to slope failures. These factors are: (i) the geological structures of a young collapse caldera with steeply dipping caldera ring faults; (ii) the geothermal system in this corner of the lake with hydrothermally altered volcanic rocks at the surface and earthquakes at 2−4 km b.s.l.; and (iii) the weather conditions in summer 2014 with sustained high temperatures and high precipitation during the days before the landslide. We describe these failure preconditions below.

### 2.1 Geological setting

The Askja volcanic system is located in the Northern Volcanic Zone of Iceland and consists of a prominent central volcano and an associated fissure swarm. The shape of the Askja central volcano is dominated by nested calderas (inset of Fig. 1). The 7–9 km wide outer caldera (Askja caldera) developed in the early Holocene, and the 3–5 km wide inner caldera (Öskjuvatn caldera) with the lake Öskjuvatn gradually subsided in the 40 years following a rifting event in 1874–1876 (Acocella et al., 2015). The ring faults of the inner caldera dissect Pleistocene glaciovolcanic deposits of the Austurfjöll and the Thorvaldsfjall mountains at the eastern and southern margin of Lake Öskjuvatn (inset of Fig. 1). There, the steep relief of up to 350 m is dominated by nearly vertically dipping fault surfaces of the cliffs and talus cones (Sigvaldason, 2002). The

caldera ring faults were the location of minor effusive eruptions in the 20[th] century, that formed, among others, the lavas Suðurbotnahraun, where the landslide originated, and Kvíslahraun in the south-eastern corner of the lake in 1922/23 (Hartley and Thordarson, 2012). The last eruptive activity at Askja occurred in 1961 when the Vikrahraun lava flowed out of a fissure at the northern rim of the Holocene Askja caldera (Thorarinsson and Sigvaldason, 1962). During the last decades, continuous

subsidence has dominated the Askja caldera (Einarsson, 1991; de Zeeuw-van Dalfsen et al., 2013), associated with contraction of an inferred shallow magma body, and the large-scale rifting of Iceland at a rate of 18.2 mm yr$^{-1}$ and an azimuth direction of 106° (DeMets et al., 1994). Nevertheless, fumarolic activity persists at the northern lakeshore in the vicinity of the Víti crater, and at the eastern and southern corner of the lake.

## 2.2 Seismicity

In the region of the Askja volcanic system, earthquakes occur at two levels in the crust, as shallow crustal seismicity between the surface and mostly 5 km depth, and as deep seismicity in the ductile lower crust at depths of 10–35 km, with magnitudes of usually $M_L < 3$ (Jakobsdóttir et al., 2002; Soosalu et al., 2010; Greenfield and White, 2015).

The deep crustal earthquakes are located in distinct regions (Fig. 1), beneath Kollóttadyngja shield volcano to the north, beneath the hyaloclastite mountain Upptyppingar to the east, location of a dyke intrusion in 2007–2008 (Jakobsdóttir et al.,

2008; White et al., 2011), at the northern part of the shield volcano Vaðalda, and beneath Askja volcano, attributed to melt migration in the lower crust (Soosalu et al., 2010; Key et al., 2011; Greenfield and White, 2015).

Shallow earthquakes cluster in the regions around the table mountain Herðubreið (Fig. 1), assigned to the differential motion of the Askja and the Kverfjöll rift segments accommodated by bookshelf faulting (Green et al., 2014) and at the south-eastern corner of the caldera lake Öskjuvatn, a region of high geothermal activity, the source location of the landslide. This

cluster at Öskjuvatn has been seen in seismic data since 1975 (Einarsson, 1991; Jakobsdóttir, 2008; Greenfield and White, 2015) and was hypothesised to be caused by hydrothermal circulation above a shallow magma body (Soosalu et al., 2010) or by thermal cracking and heat extraction in the crust (Einarsson, 1991).

Relocation of 86 earthquakes in this south-eastern hydrothermal area at lake Öskjuvatn showed that the events are concentrated at depths between 2–4 km b.s.l. (Greenfield et al., in press). Furthermore, this study showed that the

earthquakes were located along a line of 2 km length stretching from the fumarolic vents, i.e. the northern edge of the landslide source area, to the northwest. During the observation period of this study from 2009 to 2015, the earthquakes were randomly distributed in space and at depth with no clear trend over time, and the focal mechanisms did not show a distinct pattern in the years before the landslide.

## 2.3 Meteorology

The highlands of Iceland have a subarctic climate with short, cool summers and long, cold winters. In the Askja area, mean January temperatures are around -8°C and mean July temperatures are usually around 6°C (Einarsson, 1984). The Vatnajökull icecap shields the central highlands from moisture coming from the southeast and precipitation rates of 600 mm

yr$^{-1}$ are relatively low compared to the Icelandic coast (Einarsson, 1984). In winter, most precipitation falls as snow and extensive patches of snow usually last well into the summer months at the Askja central volcano. This was also the case in July 2014 (Helgason et al., 2014).

The area around the Askja central volcano experienced a period of warm weather in July 2014 with a mean monthly temperature of 8°C, 2 degrees higher than the long-term average of the mean July temperatures. In mid-July 2014, average daily temperatures ranged between 8–11°C and maximum daily temperatures were between 12–15°C (Fig. 2). The day of the landslide, 21 July, was one of the warmest of 2014 in the Icelandic highlands with temperatures around 22°C. The fair-weather period in mid-July 2014 was relatively dry but weather station Kárahnjúkar, 43 km east of Askja, recorded 9.3 mm of precipitation on 19 July and 8.7 mm on 20 July (Fig. 2). On the day of the landslide, the station recorded minor precipitation events with a total of 0.5 mm in the morning and over mid-day. These meteorological conditions with warm and wet weather in the days before the landslide increased the availability of water, also due to snow melt and rain on snow, which may have resulted in enhanced infiltration into the landslide body. This was facilitated by numerous cracks that had developed on top of the landslide body a year before the failure (Fig. 3). Higher water content increases the pore pressure and, in turn, lowers the critical stress necessary to initiate slope failures (Iverson et al., 1997; Gaucher et al., 2015).

## 3 Seismic signal analysis

A network of 58 seismic stations was in place in the Icelandic highlands clustering around the Askja volcanic system from 2009 to 2015 to investigate the crustal structure and magma migration beneath the Askja central volcano. The stations were equipped with broadband to semi-broadband seismometers of the type Güralp CMG-6TD (30 s – 100 Hz), CMG-ESPCD (60 s – 100 Hz), and CMG-3T (120 s – 100 Hz) with Nanometrics Taurus data loggers, recording at 100 Hz sampling frequency. Based on data availability, the records of 52 stations were used in this study.

Coalescence Microseismic Mapping, CMM, was used to automatically detect, locate and classify crustal earthquakes in the Askja region (Drew et al., 2013). This method combines seismic imaging and travel time inversion to determine the locations and times of earthquakes from seismic data recorded continuously on a sparse local seismometer array. Data inversion is done as a 3D subsurface grid search over the data and network of trial locations for likely locations and origin times of seismic events (Drew et al., 2013). Between 21 June and 16 August 2014, twelve events with local magnitudes $M_L$ < 2 were on average detected per day within the Askja region by the whole network. In the days before the landslide, the crustal seismicity was within this background rate and on the day of the landslide the only event detected within the Askja caldera was a $M_L$ = 0.5 earthquake at 11 km depth, 1.5 km NE of the landslide source area at 15:15:19 UTC (Fig. S1). Earthquakes occurring close to the surface and down to 5 km depth clustered in the south-eastern corner of the Askja caldera beneath the landslide source area during the weeks before and after the landslide, with about two events per day (Figs. S1 and S2).

### 3.1 High frequency seismic data analysis

To investigate the characteristics of the seismic signals of the landslide, we removed the instrument response, the mean and the trend, and band-pass filtered the signals between 1–45 Hz. We also computed spectrograms from the deconvolved East components of the seismic signals with time windows of 1.1 and 1.5 s and overlaps of 90%.

The high-amplitude short-period signals generated by the catastrophic failure part of the Askja landslide sequence can be seen in the data of all stations of the network up to a distance of 110 km (station SKAF, south of Vatnajökull glacier, see Fig. 1 for location). The seismic signal onset at the closest station MOFO, 3.5 km southeast of the landslide source area (see Fig. 1 for location), was recorded at 23:24:05 UTC (Fig. 4) and started with a smooth increase in seismic ground velocities in the first 45 s (Fig. 5). Amplitudes peaked 45 s and again 75 s after the first wave arrival with ground velocities of up to 51 µm s$^{-1}$

(Fig. 5d). Given that the amplitudes were generally higher for the horizontal components of the signal we attribute the signal to surface waves. The short-period signal lasted for about 130 s and the waveform has a symmetric, spindle-like shape (Fig. 4c). The emergent onset of the signal without clear *P* and *S* wave arrivals and no distinct peak amplitudes in the frequency bands >1 Hz is characteristic of seismic signals generated by gravitational instabilities (Suriñach et al., 2005; Deparis et al., 2008; Dammeier et al., 2011; Burtin et al., 2013).

The spectrogram of station MOFO reveals that most energy was released within the first 2 minutes of the landslide until 23:26:00 UTC, with the frequencies between 1–4 Hz containing the largest part of the seismic energy. The spectrogram of the landslide has a triangular shape where the higher frequencies decrease more rapidly in energy over time (Fig. 4a). This shape is common for landslides (Bottelin et al., 2014, Dammeier et al., 2015) and has been related to greater ground attenuation of higher frequencies and material entrainment during the propagation of the mass movement (Aki, 1980;

Surinach et al., 2005, Dammeier et al., 2011). After these 2 minutes of persistent high seismic amplitudes and energies between 1–15 Hz, amplitudes and energies decayed rapidly in the subsequent 4 minutes, followed by 10 minutes during which seismic amplitudes decreased less rapidly. Approximately 40 minutes after the end of the high amplitude signals, the background noise level was re-established.

### 3.2 Landslide force history inversion of long-period signals

Long-period seismic waves radiated by landslides result from the cycle of unloading and reloading of the solid Earth (Fukao, 1995, Takei and Kumazawa, 1994). This broad loading cycle is produced by the bulk acceleration and deceleration of the landslide mass (Okal, 1990). Long-period seismic signals (12.5–50 s, corresponding to frequencies of 0.02–0.08 Hz) were recorded for the catastrophic failure part of the Askja landslide sequence at all stations of the network (station furthest away is LAUF, at 130 km distance, located SW of Vatnajökull glacier, see Fig. 1 for location). As for the short-period signals, the

long-period waves first arrived at station MOFO at 23:24:05 UTC and lasted for approximately 130 s. The onset of these long-period waves coincided with the arrival of the short-period waves.

### 3.2.1 Method

Following the method developed by Ekström and Stark (2013) and Chao et al. (2016), we performed an inversion of the long-period landslide signals between 0.02–0.08 Hz (50–12.5 s), fitting synthetic waveforms to the data (Fig. S3) and treating the landslide mass as a single block. The selected frequency range is suitable for the inversion as higher frequencies would be affected by local-scale structures and inaccuracies in the velocity model, and lower frequencies have insufficient signal-to-noise ratios (SNR). The spatial scale of the landslide event is also small enough compared to the wavelength of the filtered seismic waves to satisfy the single block approximation (i.e. seismic point source). For the inversion, we used the 1-D velocity model of the Askja region developed by Mitchell et al. (2013) and the records of eleven broadband stations of the seismic network. We selected those stations because their data have high SNRs and they were equipped with CMG-ESPCDs or CMG-3Ts capable of recording frequencies between 0.0167−100 Hz (60–0.01 s), low enough for the long-period landslide signals. The synthetic waveforms are computed for the best-fit solution of the time-dependent forces using a signal length of 130 s, corresponding to the length of the recorded long-period signals. A recent study demonstrated that parts of the landslide material entered the lake and created a tsunami (Gylfadottir et al., 2017). The interaction between the tsunami waves and the lakeshore can contribute to the long-period seismic waves. However, we assume that the waveform data used in the inversion (Fig. S3) are directly caused by the moving mass of the landslide.

Uncertainties in the waveform inversion mainly result from the quality of the recorded data. Chao et al. (2016, 2017) demonstrated that only a few stations with good SNR are sufficient to produce reliable inversion results (i.e. a waveform fitness value larger than 0.75, a parameter that is quantified by the variance reduction and the normalised cross-correlation coefficient). In fact, the SNR of the waveforms depends on the frequency range of the band-pass filter. In order to test the sensitivity of the waveform inversion to the chosen frequency range, we tested frequency bands of 0.02–0.05 Hz (50–20 s), 0.02–0.08 Hz (50–12.5 s) and 0.04–0.08 Hz (25–12.5 s).

### 3.2.2. Results of the landslide force history inversion

Assuming a block model with a constant landslide mass over time, the inverted forces can be expressed as the product of the mass and the time-series acceleration. We obtained a maximum inverted force of $3.219 \times 10^{10}$ N and the force-time evolution for the north, east and upwards component of the loading and unloading forces (Fig. 5a). The unloading forces due to the accelerating mass of the landslide are oriented towards the SE (red arrows in Fig. 5d). In turn, the reloading forces due to the decelerating and depositing mass of the landslide strike to the NW (blue arrows in Fig. 5d). These directions are in agreement with the NW-directed propagation path of the landslide that we inferred from direct field observations of the landslide source and deposition area in August 2015 (Fig. 5e).

Assuming a block model with a constant landslide mass over time, we can estimate the acceleration time-series by dividing the resulting forces by the mass. Then, the block mass trajectories (the three-dimensional displacement time-series) can be

obtained from twice integrating the three-dimensional acceleration time-series. Here, we put the starting point at the centre of mass of the initial sliding block and find the most likely mass by ensuring that the block-mass trajectory inferred from the acceleration time-series matches the run-out path from satellite images and field observations. Recent studies suggested that the submerged landslide material travelled about 2 km at the bottom of the lake before coming to arrest (Hoskuldsson et al.,

2015; Gylfadottir et al., 2017). The parts of the landslide material hitting the water surface and slumping into the lake may have contributed to the seismic waves recorded by the broadband seismic stations. However, the material of the submerged sliding is expected to be smaller in mass than the bulk material coming to arrest on land, which might be the reason why the arrest of the submerged sliding is not recorded seismically. Thus, we used the run-out distance on land, about 1200 m, estimated from Fig. 7b, in our inversion scheme of the landslide mass. As the mass was directly inverted from the landslide

forces, smaller values of a moving mass lead to larger accelerations and longer run-out distances.

For the grid-search method of the mass computation, we used a landslide mass range between $1\times10^{10}$ kg and $2\times10^{11}$ kg. We averaged over the inversion results of the three used frequency bands taking the standard deviation into account and obtained the most likely run-out path of the landslide event with a total horizontal displacement of $1260\pm250$ m and a vertical displacement of $430\pm300$ m (Fig. 5c and e) for a resulting landslide mass of $7–16\times10^{10}$ kg. The best-fit solutions for the

synthetic seismograms have a high average waveform fitness value of 1.343 (Fig. S3).

Based on the calculated maximum force of $3.219\times10^{10}$ N and $7–16\times10^{10}$ kg of mobilised mass, the potential energy released during this landslide is estimated to be $8.2–51.5\times10^{13}$ J. Assuming an average density of 2000 kg m$^{-3}$, representative of typical values for highly fractured and hydrothermally altered Pleistocene hyaloclastites and Holocene basaltic lava flows (Moore 2001), the collapsed volume was $35–80\times10^6$ m$^3$. This volume range overlaps with prior volume estimates from field

observations and bathymetric surveys of the lake, giving $12–50\times10^6$ m$^3$ for the landslide volume (Hoskuldsson et al., 2015; Saemundsson et al., 2015; Gylfadóttir et al., 2017). Sonar investigations detected the deposits of the landslide in the lake as far as 2000 m away from the entry point of the material into the water (Hoskuldsson et al., 2015) and a calculation of the landslide volume deposited in the lake based on the rise of the water level is $10\times10^6$ m$^3$ (Gylfadóttir et al., 2017). This is less than half the total landslide volume.

In order to compare the seismically inferred run-out trajectories with the high-frequency seismic signals recorded at the closest station MOFO (Fig. 5d), we computed the travel-times of the point-source mass on the run-out path (dots shown in Fig. 5e). Notably, a late-arriving seismic phase can be observed in the high-frequency horizontal envelope waveform, which might be generated by parts of the landslide material hitting the shoreline and sliding into the lake. At this stage of the landslide history (third blue dot in Fig. 5e), we obtained an averaged maximum sliding velocity of the block mass of $7\pm0.7$ m

s$^{-1}$ from the waveform inversions of the different frequency bands (Fig. 5b). In their tsunami modelling, Gylfadottir et al. (2017) calculated the velocity of the landslide hitting the shoreline to be about 30 m s$^{-1}$, which is larger than our estimated value. We attribute this discrepancy to i) the limited applicability of a constant mass assumption in the waveform inversion, ii) the fact that the inversion gives the velocity of the total landslide mass whereas the tsunami modelling is calculating the

velocity of the front of the slide, and iii) uncertainties in the volume of the material sliding into the lake used for the tsunami modelling.

## 4 Tremor

Seismograms recorded 30 minutes before the high-energy landslide (~22:55 UTC) show gradually increasing amplitudes in the 1–45 Hz band (Fig. 6). This amplitude increase is visible on stations up to 30 km away from the landslide area. For the nearest station MOFO, the seismic amplitudes were up to three times higher than the background 7 minutes before the onset of the high-energy landslide signal (23:17 UTC, Fig. 6b). This amplitude increase was followed by an amplitude drop to values slightly below the background 2 minutes before the onset of the catastrophic part of the landslide signal (Fig. 6). We

refer to the signal with increased amplitude as seismic tremor. Here, we use the term seismic tremor to refer to any emergent, long duration seismic signal that lacks clear body wave arrivals (McNutt, 1992; Beroza and Ide, 2001), rather than to describe the source process responsible for generating seismic waves.

The observed seismic tremor initially has energy that is contained in spectral peaks centred at 2.3, 4.3, and 7.1 Hz (Fig. 6c). Tremor with a sharply peaked spectrum consisting of a fundamental frequency with overtones is called harmonic tremor.

The Askja tremor is therefore approximately – but not perfectly – harmonic. Furthermore, harmonic tremor that gradually evolves in time is said to be gliding (McNutt, 2005). A particularly eye-catching aspect of the Askja tremor is the concurrence of up-gliding and down-gliding spectral lines (Fig. 6a). Specifically, at about 23:14 UTC, the spectral content of the tremor started to change and both up and down gliding frequency bands can be observed simultaneously (Fig. 6c). Tremor amplitudes are higher for the horizontal components than for the vertical component, as it is the case for the

landslide signal. In contrast to the signal of the landslide, which that also contains long-period seismic waves, the tremor is confined to frequencies above 1 Hz. Contemporaneously with the amplitude drop, the gliding spectral lines stopped at about 23:22 UTC and a period of 2 minutes of quiescence can be seen in the spectrograms before the high-energy signal of the catastrophic landslide starts.

We begin in Section 4.1 by discussing the location of the seismic tremor. In Section 4.2, we present numerical simulations of

a particular tremor-generating process: the repeated stick-slip motion of a small region along the landslide failure plane. We emphasize that repeated stick-slip motion is only one possible explanation of seismic tremor. Other possible explanations are discussed in Section 6.2. In Section 6.2, we conclude that repeated stick-slip motion is the most likely tremor-generating process at Askja, while keeping in mind that improved observations in the future would be useful to offer more clear evidence of the tremor-generating process. We furthermore note that the Askja tremor is a rich and intricate seismic signal;

here we focus on its most robust and clear aspects.

## 4.1 Tremor location

For a rough estimation of the tremor location and to check whether it is not only temporally but also spatially correlated with the landslide, we computed the ratios of the mean envelope amplitudes of 1 minute of the tremor to 3 minutes of background seismic noise for all stations of the network. First, we removed the instrument response, the mean and the trend, and band-pass filtered the signals between 1–45 Hz. Then, we computed the envelopes for 1 minute of the tremor starting at 23:17:00 UTC, 21 July 2014, and for 3 minutes of background seismic noise starting at 00:10:00 UTC of the same day for the E components. Next, we calculated the mean amplitudes of the envelopes for these two time windows and determined their ratio. The mean envelope amplitude ratio is highest, up to 3.2, at the stations closest to the source area of the landslide and decays to values of 1 for stations tens of kilometres away from Lake Öskjuvatn (Fig. S4). However, we note that the decrease of the mean envelope amplitude ratio has an elliptical outline with a long axis oriented NE-SW, parallel to the orientation of the general structural trends at the Askja volcanic system (Fig. 1), which are probably responsible for seismic wave attenuation effects.

To further refine the location of the tremor, we used the procedure of Burtin et al. (2013) to locate the tremor signal on a DEM grid. This statistical approach assigns a probability of being the source of the signal to each grid point based on cross-correlation of the signal envelopes of different stations. The resulting probability density function is normalised to its maximum value giving this grid point a likelihood of 1 to be the source location of the signal (Burtin et al., 2014). We worked with the data of 21 stations for the location that showed the gliding spectral lines in the spectrograms, and with a DEM with a grid spacing of 100×100 m. We used a frequency range of 1.5–3 Hz as this frequency band shows the highest tremor energy, and time windows of 1 minute starting at 22:54:00 UTC. With this location method, we found that the tremor signal was most likely located at the south-eastern shore of the caldera lake at Askja, where fumaroles are the surface expression of the hydrothermal system (Fig. 7a). This is the northern corner of the landslide source area. Over 30 minutes before the landslide, the likely tremor location only changed by a few 100 m. We tested the influence of the seismic wave velocity on the results by varying this parameter in the location routine between 500 and 3700 m s$^{-1}$. The best-fit locations for the different wave velocities differ up to 500 m from each other but remain at the south-eastern lakeshore.

## 4.2 Numerical simulations of seismic tremor

### 4.2.1 Method

To investigate further the seismic tremor observed before the Askja landslide, we conduct numerical simulations of stick-slip motion and elastic wave propagation. Our approach calculates the force balance between elastic stresses, including elastic wave propagation, and an interface strength set by rate-and-state friction. More details about these simulations are given by Lipovsky and Dunham (2016).

The aseismic-seismic transition is a central feature of sliding under rate-and-state friction (Rice et al., 2001). This transition is commonly expressed as a critical patch size $R_c$, defined such that − with all other parameters held constant − a given interface will experience stick-slip oscillations if $R > R_c$, with $R_c$ defined as

$$R_c = \frac{d_c G}{(b-a)\sigma - \eta\, v_0} \tag{1}.$$

In this expression, $\sigma$ is the effective normal stress, $G$ is the shear modulus, $d_c$ is the frictional state evolution distance, $a$ is the magnitude of transient peak strengthening during step loading, $b$ is the magnitude of strength change between peak strength and steady state, $\eta = \rho c_s$ is the shear wave impedance with density $\rho$ and shear wave speed $c_s$, and $v_0$ is the nominal loading velocity. The parameter ($b$-$a$) must be positive for stick-slip cycles to occur; an interface with this property is called rate weakening. Frictional parameters are taken from laboratory experiments (Marone, 1998) and we use typical values for

crustal rocks (Table 1). The interface normal stress must be prescribed, and for this value we use an overburden stress calculated from a landslide thickness of 30 m, consistent with previous work (Gylfadóttir et al., 2017).

Under rate-and-state friction, a change in the repeat time $T$ of the stick-slip events may occur for a number of reasons. Near the transition between steady and stick-slip sliding, $T$ scales approximately as

$$\frac{T}{T_c} = \frac{R}{R_c} \tag{2}$$

where $T_c$ is the lowest achievable repeat time (Lipovsky and Dunham, 2017)

$$T_c = 2\pi\sqrt{\frac{a}{|a-b|}\frac{d_c}{v_0}} \tag{3}.$$

### 4.2.2 Results of the tremor simulations

By matching synthetic and observed seismograms, we are able to explain two prominent observations (Fig. 8). First, we find that the gliding of the spectral tremor lines can be produced by stick-slip earthquakes occurring with changing frequency.

Second, we reproduce the aseismic period immediately before the main landslide failure. As both up- and down-gliding spectral lines occur simultaneously in the Askja dataset, we infer that more than one source was active at the same time, each producing tremor. Hence, we use two simulations. We emphasize however, that insofar as the resulting tremor simulations qualitatively resemble the observed tremor, the parameterisation that we have chosen is highly non-unique. Other parameterisations are possible, and we focus here on the following parameterisations simply for the purpose of

demonstrating that repeated stick-slip motion of a region of the failure plane was a likely tremor-generating process before the Askja landslide.

In the first simulation, by increasing the initial loading velocity $v_0$ = 0.6 mm s[-1] by 0.01 (mm s[-1]) min[-1] (see Tab. 1 for the simulation parameters), the repeat time $T$ between the stick-slip events decreases and the synthetic spectrogram shows up-gliding spectral lines with a fundamental frequency of 2.5 Hz and overtones of 5 Hz and 7.5 Hz (Fig. 8c). The spectral lines

contain less energy with time and fade at 13 minutes.

In the second simulation, the repeat time $T$ between the stick-slip events increases with time and downward spectral gliding

can be seen in the synthetic spectrograms. The increase in $T$ can be achieved by a deceleration in loading velocity or by an expanding stick-slip region. We elaborate on these possibilities in the discussion. Here, we report that simulations with a patch radius of $R = 30$ m that grows by 10 mm s$^{-1}$ show spectral lines starting at frequencies of 4 Hz, 8 Hz and 12 Hz and gliding down to frequencies of 3 Hz, 6 Hz and 9 Hz before abruptly disappearing at 12 minutes (Fig. 8d).

Although the stick-slip simulations can reproduce the fundamental frequency and some overtones of the observed tremor we acknowledge that some overtones of the simulations are not clearly visible in the data. The overtone labelled with number two (Fig. 8b and c) is less strongly observed than others, for example. We believe that the simplest explanation for this is that our basic model of wave propagation fails to account for certain propagation phenomena that may diminish wave amplitudes. Wave propagation in the complicated, 3D, layered, attenuating media surrounding the Askja volcanic complex is

far richer than we have attempted to capture.

We calculated the stress drop $\Delta\tau$ in each small, repeating stick-slip event as

$$\Delta\tau = \alpha G \frac{u}{R} \tag{5}$$

where $\alpha$ is a geometrical constant usually taken to be $7/16\ \pi \sim 1.37$, and u is the slip in each event. With the parameters of our best-fit model, u=0.24 mm, R=30 m, and G=7 GPa, we calculated a stress drop of 77 kPa. The scalar moment is

$M_0=4.75\times10^9$ Nm, which is equivalent to a moment magnitude $M_w = 0.42$.

## 5 Afterslides

During 8 hours after the main landslide, several other high-amplitude short-period signals of much lower amplitude were recorded (for example, at 23:41:05 UTC, Fig. 4d). Their waveforms are spindle-shaped with dominant frequencies of about 1–2 Hz. The signals are only visible at frequencies >1 Hz and a force history inversion of low-frequency signals is not

possible. These events lasted between a few seconds and a minute and have characteristics such as emergent onsets, slowly decaying tails, and triangularly shaped spectrograms that are indicative for slope failures (Norris, 1994; Dammeier et al., 2011; Burtin et al., 2013; Chen et al., 2013). We attribute these signals to smaller slope failures that occurred after the main landslide. Following the nomenclature for earthquakes with the main shock and subsequent, smaller aftershocks happening in the same area, we here introduce the term afterslides for smaller mass movements occurring after a large landslide on the

same landslide scar and in its close vicinity.

In the first and second hour after the landslide, eleven and seven afterslide events were recorded, respectively (Fig. 9a). Afterwards, the amplitude, energy and frequency of the afterslides decreased gradually. We observed 2–4 events per hour for the third to the eighth hour after the main landslide. After eight hours, no more afterslides were detected. To locate the seismic signals of the afterslides on a 20×20 m DEM grid, we used the same location method of Burtin et al. (2013) that we

applied for the tremor localisation. In the first hours after the main landslide, the afterslides originated in the source area of the main slide and along the caldera ring fault at the south-eastern side of Lake Öskjuvatn (Fig. 9b). Later afterslides tended to cluster at the top part of the destabilised walls from which the main landslide detached.

## 6 Discussion

### 6.1 Dynamics of the landslide sequence from high- and low-frequency signals

Combining seismic data analysis and field observations reported in the literature and made during a field campaign in August 2015, we are able to summarise the factors that lead to the landslide and describe the precursory tremor, the landslide and the subsequent small slope failures in detail.

Crack opening started years before the landslide at the head wall of the slide as documented in pictures taken from 2011 onward (Helgason et al., 2014). These cracks helped in dissociating the landslide from the surrounding ground mass. The warm weather with a number of precipitation events in July 2014 further promoted crack opening by bringing moisture to the Askja caldera and increasing the snowmelt, both giving rise to higher pore pressure. On 21 July 2014, at about 22:55 UTC, that is half an hour before the main catastrophic landslide failure, a complex harmonic tremor signal with a fundamental frequency of 2.3 Hz and several overtones emerged from the background noise in the seismic data, which we interpret as the start of the detectable slow downslope movement of the landslide mass. The spectral lines of the tremor signal changed their frequency content during an 8 minute period starting at 23:14 UTC. Synchronous up- and down-gliding of the frequency bands could indicate that several sliding planes at the base of the landslide experienced stick-slip motion at the same time. As the waveforms of the stick-slip earthquakes have to be similar for their merged signal to be visible as approximately harmonic tremor with overtones we envisage that this happens because the moving patches gradually slide over asperities at their base.

Through acceleration and growth of the sliding planes, the stick-slip sliding transitioned into an aseismic, stable sliding period 2 minutes before the bulk landslide mass failed catastrophically. Based on combined inspection of the high- and low-frequency signals generated by the Askja landslide, we distinguish three phases of landslide motion, initiation, propagation and termination as proposed by Hibert et al. (2014) and Chao et al. (2016). The initiation phase of the landslide started immediately before high-amplitude surface waves arrived at the nearest station at 23:24:05 UTC. The landslide force history inversion shows a sharp increase in the accelerating force during the first 30 s of the landslide signal (Fig. 5a), generated by the onset of motion of the landslide's bulk mass. The high-frequency signals show an emergent onset during these first 30 s of fast landslide motion (Fig.5d). These signals reach maximum amplitudes about 45 s after the signal onset, which coincides with lower acceleration and the transition to decelerated motion in the force history inversion of the propagation phase. We infer from this lag time in the high-frequency signal that the main slope failure along the south-eastern caldera wall was a single, large event, starting with aseismic sliding of a relatively coherent mass that gradually fragmented during down-slope acceleration (cf. Allstadt, 2013; Hibert et al., 2015). In this interpretation, the high-frequency signals are caused by the momentum exchanges of block impacts, and frictional processes within the moving slide and along its boundaries, especially when the moving mass traverses small-scale topographic features on the sliding base (cf. Dammeier et al., 2011; Allstadt, 2013). These multiple sources, along with the diversity of propagating waves, were responsible for the multiple amplitude pulses and the lack of a clear maximum of the seismic amplitudes in the higher frequencies (Deparis et al., 2008; Dammeier

et al., 2011). The deceleration phase of the landslide lasted for about 70 s (Fig. 5b), a period during which the high-frequency amplitudes also gradually decline (Fig 5d). This termination phase of the landslide was associated with material deposition at the shore but also into Lake Öskjuvatn NW of the landslide source area.

From the landslide force history inversion, we calculate that a total mass of $30–80\times10^6$ m$^3$ of hyaloclastic material was involved in the slide and a fraction of $10\times10^6$ m$^3$ entered Lake Öskjuvatn creating a tsunami (Gylfadóttir et al., 2017). As a result of the removal of overlying mass, the hydrothermal system below the landslide source area was depressurised and a cloud of steam and landslide dust rose above the caldera (Helgason et al., 2014).

During the 8 hours after the main landslide, subsequent small slope failures occurred at the destabilised caldera walls. The rolling, jumping, colliding and impacting blocks created seismic signals with emergent onsets, and spindle-shaped envelopes (Dammeier et al., 2011; Allstadt, 2013; Hibert et al., 2015; Moretti et al., 2015) and with higher seismic amplitudes than the background level at the stations closest to the Askja caldera. Such a chain-reaction with subsequent slope collapses is not uncommon after landslides (Iverson et al., 2015). Similar to earthquakes and their aftershocks that occur less frequently and with smaller amplitudes with time after the main shock (Omori, 1894, Gutenberg and Richter, 1956), we observe a decay in the size and frequency of the small slope failures following the main landslide that we call afterslides.

## 6.2 Source process of the seismic tremor

Seismic tremor has been observed in a variety of settings including tectonic subduction zones, volcanoes, subsurface reservoirs, glaciers, ice sheets, and landslides. Reflecting these diverse settings, an equally diverse collection of physical processes may explain the source process responsible for creating seismic tremor. Possible sources of seismic tremor include: (i) fluid-flow-induced oscillations of conduit or fracture walls (Julian, 1994; Hellweg, 2000; Rust et al., 2008; Matoza et al., 2010; Corona-Romero et al., 2012; Dunham and Ogden, 2012; Unglert and Jellinek 2015); (ii) resonance of fluid-filled cracks or pipes with open or closed ends (Chouet, 1985, 1986, 1988; Benoit and McNutt, 1997; Jousset et al., 2003; Neuberg, 2006; Jellinek and Bercovici, 2011; Röösli et al., 2014; Sturton and Walter et al., 2015; Lipovsky and Dunham, 2015); (iii) bubble growth or collapse due to hydrothermal boiling of groundwater (Leet, 1988; Kedar et al., 1998; Cannata et al., 2010); and (iv) continuously repeating processes such as stick-slip motion (Neuberg, 2000; Powell and Neuberg, 2003; Dmitrieva et al., 2013; Hotovec et al., 2013; Lipovsky and Dunham, 2016; see also reviews by McNutt, 1992 and Konstantinou and Schlindwein, 2003). We note that the first three of these processes are hydraulic in origin.

Mechanical analyses of hydraulic sources for seismic tremor showed that fluid-flow instabilities producing wall oscillations (Julian, 1994) require flow speeds on the order of the speed of sound (Dunham and Ogden, 2012), thus suggesting that the applicability of these physics is limited to situations such as high velocity volcanic jets. As the Askja landslide was not associated with any volcanic activity that would support this mechanical model of tremor generation through hydraulic processes, we conclude that a hydraulic source is unlikely to explain the phenomena observed at Askja.

Furthermore, Lipovsky and Dunham (2015) analysed seismic tremor due to hydraulic resonance and found that the resonant frequencies of a hydraulic fracture are expected to be unevenly spaced following $f_n/f_1 = n^{3/2}$. Complementary, Lipovsky and Dunham (2016) showed that a simple application of the Fourier transform to a repeating sequence of slip pulses results in a frequency pattern of $f_n/f_1 = n$. For a fundamental tone of $f_1 = 2.3$ Hz as observed for the Askja landslide tremor, we would

expect its first harmonic at 6.5 Hz for a resonating fracture or at 4.6 Hz for a stick-slip source. Observations of the Askja landslide tremor show that the spectral peaks are relatively evenly spaced with $f_1 = 2.3$ Hz, $f_2 = 4.3$ Hz, and $f_3 = 7.1$ Hz, a pattern that is in closer agreement with the harmonic relationship $f_n/f_1 = n$. This provides observational evidence for a stick-slip mechanism and against a hydraulic source of the tremor before the Askja landslide.

Several additional lines of reasoning support the interpretation of the Askja landslide tremor as being due to repeating stick-

slip motion along the landslide failure plane. Stick-slip montion have been observed as precursors to other landslides (Yamada et al., 2016; Poli, 2017), although in these cases individual stick-slip events could easily be distinguished. Although individual events are roughly discernible in seismograms, the effects of attenuation and superposition of multiple sources makes time domain analysis difficult (Fig. S7). Following previous studies, we therefore prefer spectral analysis over the time domain (Dmitrieva et al., 2013; Hotovec at al., 2013; Winberry et al., 2013; Lipovsky and Dunham, 2016). We note that

the tremor observed by Yamada et al. (2016) was observed at a much shorter source-to-station distance <1 km, whereas our closest station is 3.5 km from the landslide source area. It is also possible that individual stick-slip events were more clearly visible in the studies by Yamada et al. (2016) and Poli (2017) because the events were either larger or were more energetic. We note that stick-slip motion has previously been proposed to cause seismic tremor on the sliding planes of sudden surface mass movements including ice-rock avalanches (Caplan-Auerbach et al., 2004, Huggel et al., 2008) and during glacier

sliding (Caplan-Auerbach and Huggel, 2007; Winberry et al., 2013; Allstadt and Malone, 2014; Helmstetter et al., 2015; Lipovsky and Dunham, 2016).

When tremor occurs due to repeating stick-slip cycles, gliding of the frequency bands is the result of a changing recurrence time (Lockner et al., 1991; Neuberg 2000; Dmitrieva et al., 2013; Hotovec et al., 2013; Lipovsky and Dunham, 2016). In Section 4.2, we demonstrated this phenomenon using a simplified numerical simulation of stick-slip motion. We are able to

produce the up-gliding spectral lines in our model by increasing the loading velocity $v_0$ (Fig. 8c). The down-gliding spectral lines can be simulated in two ways: (i) by decreasing the loading velocity or (ii) by a growing stick-slip patch. Given the ensuing landslide motion, deceleration of a patch would only be realistic with a subsequent accelerating patch taking over the momentum. However, the observations show up-gliding spectral lines, the expression of an accelerating tremor patch, before the down-gliding spectral lines (Fig. 8a, b). Therefore, we find the explanation of a decelerating patch to be physically

unrealistic and prefer to attribute downward gliding spectral lines to an expanding stick-slip region. Changes in the stick-slip patch size can also explain the several minutes of increasing tremor amplitudes (Fig. 6b) as being due to a proportional increase in the moment release in each stick slip cycle. Observed seismic amplitudes increased by a factor of three over seven minutes (Figs. 6 and S4), which would correspond to an increase in patch dimension by a factor of $\sqrt{3}$. If the initial

patch radius was 30 m (as fits the data from the up-gliding patch), then this corresponds to an average radial growth rate of 70 mm/s.

Our stick-slip simulations additionally predict the disappearance of the tremor signal shortly before the landslide at different times in the simulations. We suggest that two different mechanisms are responsible for this behaviour in our case because we assume that the two stick-slip tremor patches move independently and hence transition into a state of seismically non-detectable movement due to different reasons. First, the patch that experiences accelerated loading eventually crosses the stability threshold and begins to slide stably ($R < R_c$ in Eq. 1). This behaviour is consistent with the theoretical prediction of a transition from stick-slip to stable sliding at high loading rates (Rice et al., 2001; Gomberg et al., 2011). In the simulations, this can be traced by the up-gliding spectral lines whose energy contents decrease with time until they fade into the background at 13 minutes (Fig. 8c). Second, the patch with a growing area experiences a commensurate increase in recurrence time (recurrence time and patch size are proportional, see Eq. 2); eventually the recurrence time becomes so large that a quiescent period ensues. This can be seen in the simulations of the down-gliding spectral lines that disappear at 12 minutes (Fig. 8d).

To further gain insight into the nature of the tremor, we stacked the signals of the eight closest stations operating at the occurrence time of the tremor (DREK, GODA, HOTT, JONS, KLUR, MOFO, STAM, VADA, see inset of Fig. 7 for locations, and Figs. S5 and S6 in the supplement for the stacked and single-station spectrograms) by first computing the spectrograms with the same specifications for each station, like time window length and fraction of window overlap, etc. Then, we added the spectrograms' energy values per frequency and time step, before dividing these sums by the number of stations to obtain the mean energies for the frequencies and time period of interest. The result shows that the gliding spectral lines of the tremor are clearly visible as sharp bands of higher energy values and did not become blurred in the stacked spectrogram (Fig. S5). The fundamental frequency of the tremor is the same, 2.3 Hz, for the stacked spectrogram as for the single station spectrograms. The maximum standard deviation of the fundamental frequency in the single spectrograms to the stacked spectrogram is 0.3 Hz. This confirms that the gliding tremor comprises the same frequencies at the tested stations. Hence, we conclude that the nature of the gliding tremor signal is a source property rather than a site or wave propagation effect.

To conclude, we propose that the Askja seismic tremor is most likely caused by repeated stick-slip motion on small, frictionally unstable patches along the landslide failure plane. This interpretation is, however, to some degree uncertain. Previous studies that interpreted gliding tremor as being due to repeating stick-slip were based on much clearer spectral signatures, often accompanied by individually discernable events (Dmitrieva et al., 2013; Hotovec at al., 2013; Winberry et al., 2013; Lipovsky and Dunham, 2016; Yamada et al., 2016; Poli, 2017). We nevertheless note that the occurrence of accelerating stick-slip motion is consistent with the onset of a large landslide. This interpretation implies that the landslide mass had already started to move before the high-energy signals of the catastrophic part of the landslide emerged in the seismic data. We envision the stick-slip patches to be located at the base of the landslide, developing along heterogeneities such as the lithological contact between the hyalocastites and the 1923 Suðurbotnahraun lavas, and pre-existing material

heterogeneities within the hydrothermally altered hyaloclastites (Fig. 7b). Stick-slip sliding taking place at the base of the landslide rather than predominately within a highly damaged rock mass would result in a better coupling and thus higher energy transmission to the ground. This explains why the tremor can be observed over 30 km away from the landslide source region. In addition, we note the observation that cracks in the head wall of the landslide started to open after 2011 (Helgason

et al., 2014) and that numerous cracks had developed at the surface of the landslide body a year before the failure (Fig. 3). This implies that the failure planes bounding the landslide developed years before the bulk movement of the landslide mass and just needed to be activated. The warm and wet weather, promoting pore pressure increase in July 2014 may have played an important role in this. Slight increases in pore water pressure can induce stick-slip motion, as has been observed on blocks of a seasonally active landslide in the French Alps (Genuchten and Rijke, 1989).

## 6.3 Tremor and rapid stick-slip as early-warning signs of landslide failure

The risk to human life posed by landslides compels us to explore the possibility of designing a landslide early-warning system based on the existence of precursory seismic tremor. Because seismometers may be placed at a distance from the landslide site, such a system would provide safety benefits compared to other types of monitoring. While landslide early-

warning systems may not be possible at the present time, our goal here is simply to outline several scientific and engineering considerations for such a system.

First, future observations should be made to determine whether accelerating stick-slip, manifested as either isolated events (e.g., Yamada et al., 2016; Poli, 2017) or as seismic tremor (e.g., as before the Askja landslide), is in fact a sufficiently common precursor to large scale slope failures. There is evidence that this may be the case. Many voluminous slope failures

do start as slow-moving landslides (Palmer, 2017) whose early motion should produce a seismic signal. Furthermore, some already-monitored slow-moving landslides show displacement rates that scale with the seismicity rates of cracks and stick-slip tremor signals (Tonnellier et al., 2013, Vouillamoz et al., 2017) and could serve as test sites.

Second, any seismicity-based landslide early-warning system will require seismic data to be analysed in near-real time by a fast and reliable algorithm. Early-warning systems for tectonic earthquakes with simpler seismic signals than slope failures

have been designed that meet this standard (Allen et al., 2009; Cua et al., 2009). Machine-learning methods could form the basis for such an algorithm as they are a powerful and promising tool to detect and classify signal classes, also of precursory slope activity, in seismic data (Hammer et al., 2012; Esposito et al., 2013; Zeckra et al., 2015). Other anticipative signals of natural gravity-driven instabilities such as those of cracking could also be detected and identified in this way. Cracking and fracturing signals have been identified in seismic data before cliff collapses (Amitrano et al., 2005; Zeckra et al., 2015),

slope instabilities (Sima, 1986; Kilburn and Petley, 2003; Kolesnikov et al., 2003; Dixon et al., 2015; Faillettaz et al., 2016; Yamada et al., 2016), and break-off of hanging glaciers (see review by Faillettaz et al., 2015).

Third, seismic networks must be able to observe landslide-related seismicity. In the case of Askja, a well-positioned network of seismic stations located a few kilometres away from the slope instability was able to detect precursory tremors. Further

study will be required to test the detection thresholds of seismic networks as a function of network design parameters including station spacing and sampling rate. Regional-scale landslide monitoring with a seismic network has only been attempted on a few occasions (Burtin et al., 2013; Hibert et al., 2014) and the challenge persists to detect landslide signals in a continuous seismic data stream in near-real time (Dammeier et al., 2016; Manconi et al., 2016; Chao et al., 2017).

**7 Conclusions**

We analysed seismic data from a voluminous landslide, its precursory tremor and successively following small slides that all occurred at the south-eastern shore of the caldera lake Öskjuvatn of the Askja central volcano in the Icelandic Highlands on 21 July 2014. The seismic data is of exceptionally high quality because (i) the 58 stations were centred around the Askja caldera, and (ii) anthropogenic noise sources are far away. We performed a detailed analysis of the seismic data that showed

that the short-period signals of the landslide mainly consist of surface waves, which arrived at the closest station at 23:24:05 UTC and lasted for about 130 s. The seismic signal of the Askja landslide is characteristic of voluminous slope failures with an emergent onset without clear $P$ and $S$ wave arrivals and a spindle-shaped envelope. Inversion of the long-period signals of the landslide reveals that the bulk mass of $30–80\times10^6$ m$^3$ propagated to the northwest starting at the caldera ring fault at the south-eastern shore of Lake Öskjuvatn, which is consistent with field observations. Subsequent small slope failures, that we

call afterslides, occur in the hours after the main landslide at the destabilised caldera walls.

We detected approximately harmonic tremor with a fundamental frequency of 2.3 Hz commencing about 30 minutes before the landslide and diminishing into 2 minutes of seismic quiescence before the catastrophic failure. By numerically simulating stick-slip motion and elastic wave propagation, we were able to reproduce the aseismic period and the simultaneously up- and down-gliding of the spectral tremor lines with models where stick-slip earthquakes occur with changing frequency. We

propose that upward spectral gliding occurs because of an increase in the recurrence frequency of stick-slip events on an accelerating sliding patch. In contrast, we explain downward spectral gliding by an expanding stick-slip region where the recurrence frequency of stick-slip earthquakes decreases. The transition from stick-slip to stable sliding is marked by a seismically quiet period of 2 minutes before the bulk landslide mass failed catastrophically. Although there is both uncertainty and non-uniqueness associated with our interpretation of the precursory seismic tremor, we argue that such a

model is the only tremor-generating process for which we have a physics-based model that is able to match observations to the degree that we have done here. We emphasise the utility of seismic networks to detect and characterise not only landslides but also the precursory signals that might otherwise go unnoticed. This is of utmost importance for sites with a high hazard potential and encourages the development of early-warning systems based on seismic data for monitoring slope failures.

**Data availability**

The seismic dataset is available upon request from Prof. Robert White, Bullard Laboratories, University of Cambridge, Cambridge CB3 0EZ, United Kingdom.

**Acknowledgements**

Seismometers were provided by the Natural Environmental Research Council (NERC) SEIS-UK under loans 968 and 1022. We would like to thank the Icelandic Meteorological Office for making the weather data available. The field campaign in summer 2015 was financially supported by an expedition fund of the Helmholtz Centre Potsdam GFZ and T. Witt, T. Walter, D. Müller, B. Steinke, and Á. Höskuldsson are thanked for their support in the field. B. P. Lipovsky was supported by a postdoctoral fellowship from the Department of Earth and Planetary Sciences at Harvard University. Some figures were

created with the help of GMT, Generic Mapping Tools, developed by Wessel et al. (2013). Eva P. S. Eibl is thanked for fruitful discussions. We gratefully acknowledge J. Caplan-Auerbach and an anonymous referee for their constructive reviews, which helped to improve this manuscript. We also thank the associate editor, Kate Allstadt, for the helpful comments and for carefully handling this submission.

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

**Table 1**

**Parameters of the stick-slip simulations**

| Parameter | Symbol | Value |
|---|---|---|
| Epicentral distance | L | 3.5 km |
| Quality factor | Q | 25 |
| Shear wave speed in rock | $c_s$ | 1878 m s$^{-1}$ |
| Density of rock | $\rho$ | 2000 kg m$^{-3}$ |
| Thickness of landslide | H | 30 m |
| Frictional state evolution distance | $d_c$ | $15 \times 10^{-6}$ m |
| Frictional direct effect parameter | a | 0.03 |
| Frictional ageing effect parameter | b | 0.04 |
| Static coefficient of friction | $\mu_0$ | 0.7 |
| Initial loading velocity | $v_0$ | 0.6 mm s$^{-1}$ |
| Repeating earthquake patch radius | R | 30 m |
| Creep acceleration on the accelerating patch (only on up-gliding patch) | $\dot{v}$ | 0.01 (mm s$^{-1}$) min$^{-1}$ |
| Rate of patch size change on the decelerating patch (only on down-gliding patch) | $\dot{R}$ | 10 mm s$^{-1}$ |

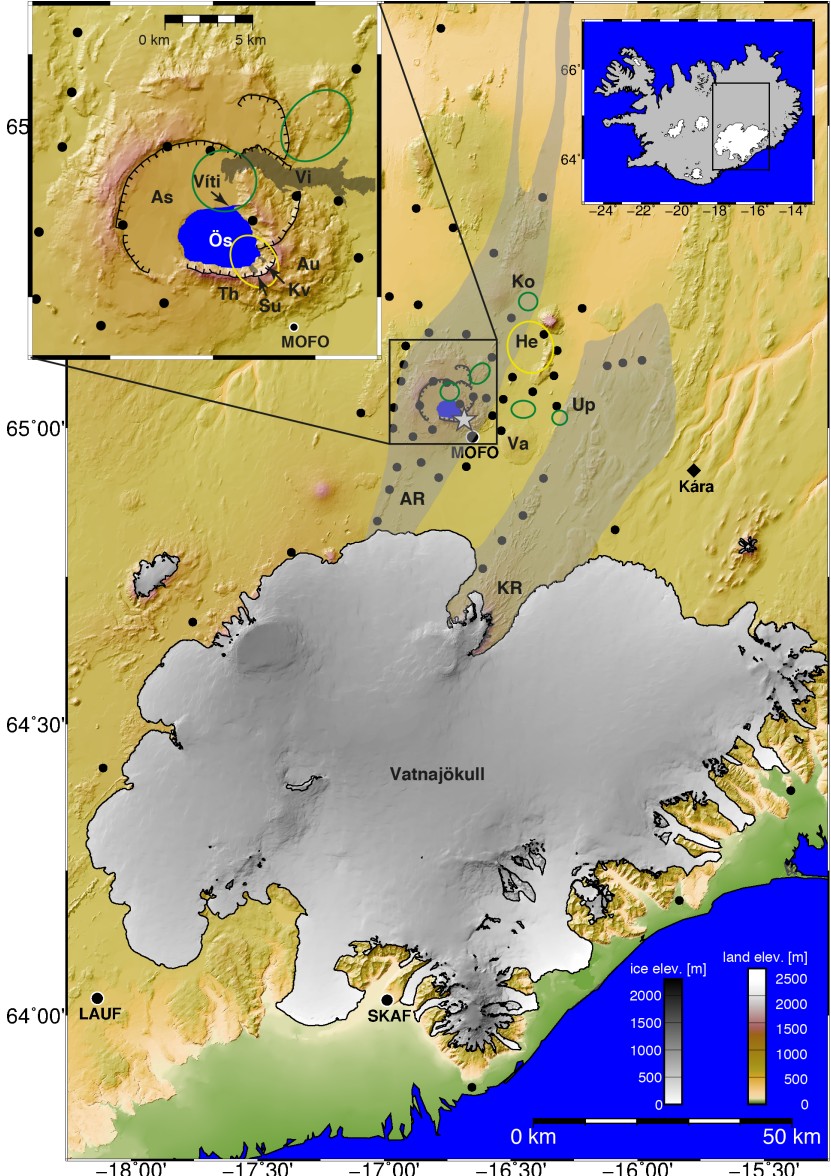

**Figure 1:** The Askja central volcano north of the ice cap Vatnajökull in the Icelandic highlands with locations of the seismic stations (black circles). The white star is the source location of the 21 July 2014 landslide. The weather station Kárahnjúkar (Kára) is indicated with a black diamond. Inset of the Askja central volcano shows the hyaloclastite mountains of Austurfjöll (Au) and Thorvaldsfjall (Th), the caldera ring faults of the Askja caldera (As) and the Öskjuvatn caldera (Ös), the small explosion crater Víti, the basaltic 1922/23 eruption sites of Suðurbotnahraun (Su) and Kvíslahraun (Kv), and the 1961 Vikrahraun (Vi) lava flow (grey shading). The areas of deep crustal earthquakes at Kollóttadyngja (Ko) shield volcano, at the hyaloclastite mountain Upptyppingar (Up), north of the shield volcano Vaðalda (Va), and at the Askja caldera north side are indicated with green ellipses. Areas of shallow crustal seismicity at the table mountain Herðubreið (He), and at the south-eastern edge of the Öskjuvatn caldera are shown with yellow ellipses. The Askja (AR) and the Kverfjöll rift (KR) segments are shaded in light grey (Einarsson and Sæmundsson, 1987).

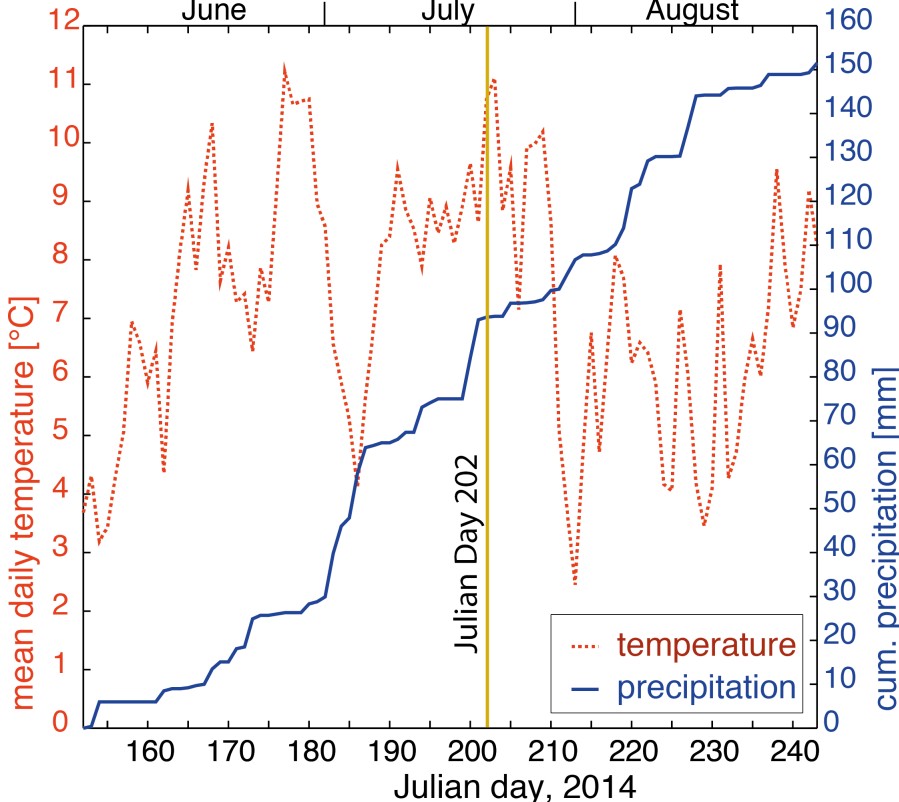

Figure 2: Mean daily temperatures and cumulative precipitation at weather station Kárahnjúkar (see Fig. 1 for location) in June, July and August 2014. Note the two days with high precipitation immediately before and the increased temperature at the day of the landslide (Julian Day 202, yellow line).

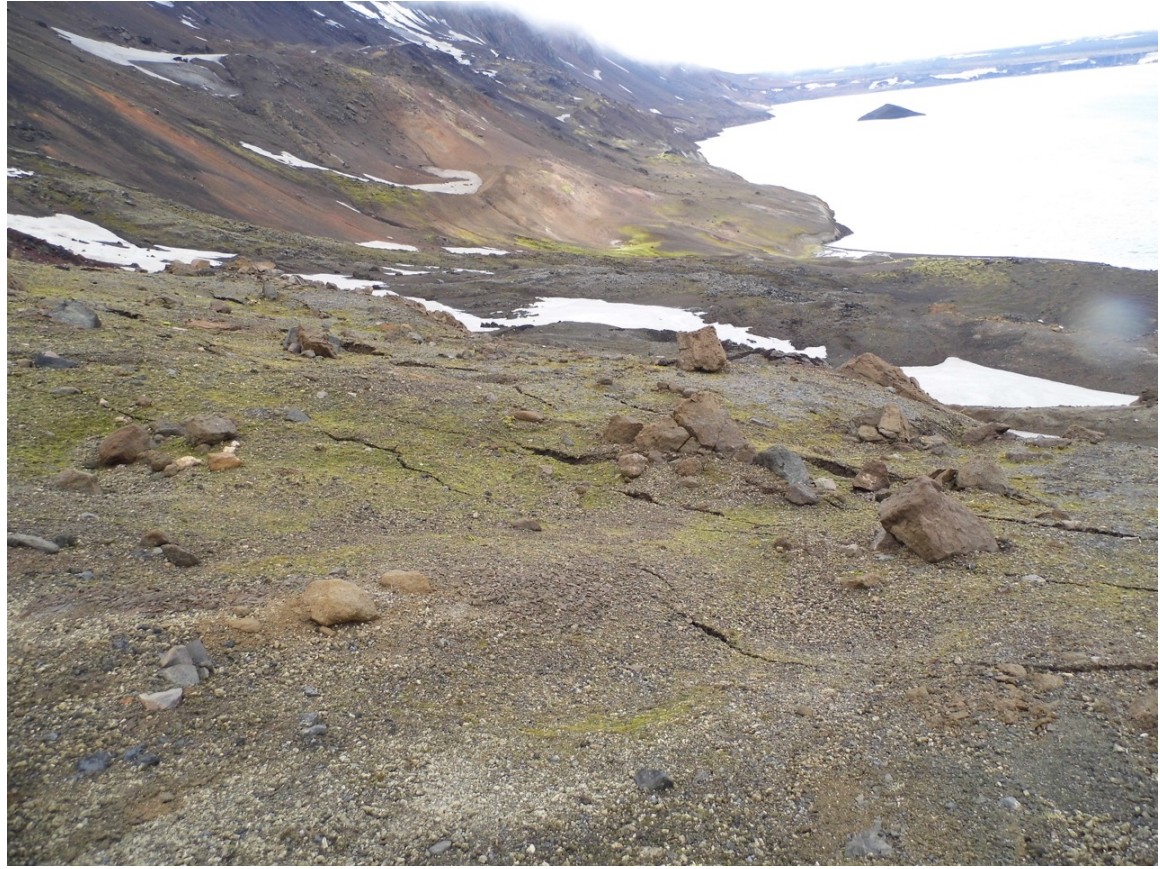

**Figure 3: Surface opening cracks on top of the landslide body in August 2013, a year before the failure. Location of the image is indicated on Fig. 5e, view is to the west. Image taken by Daniele Trippanera.**

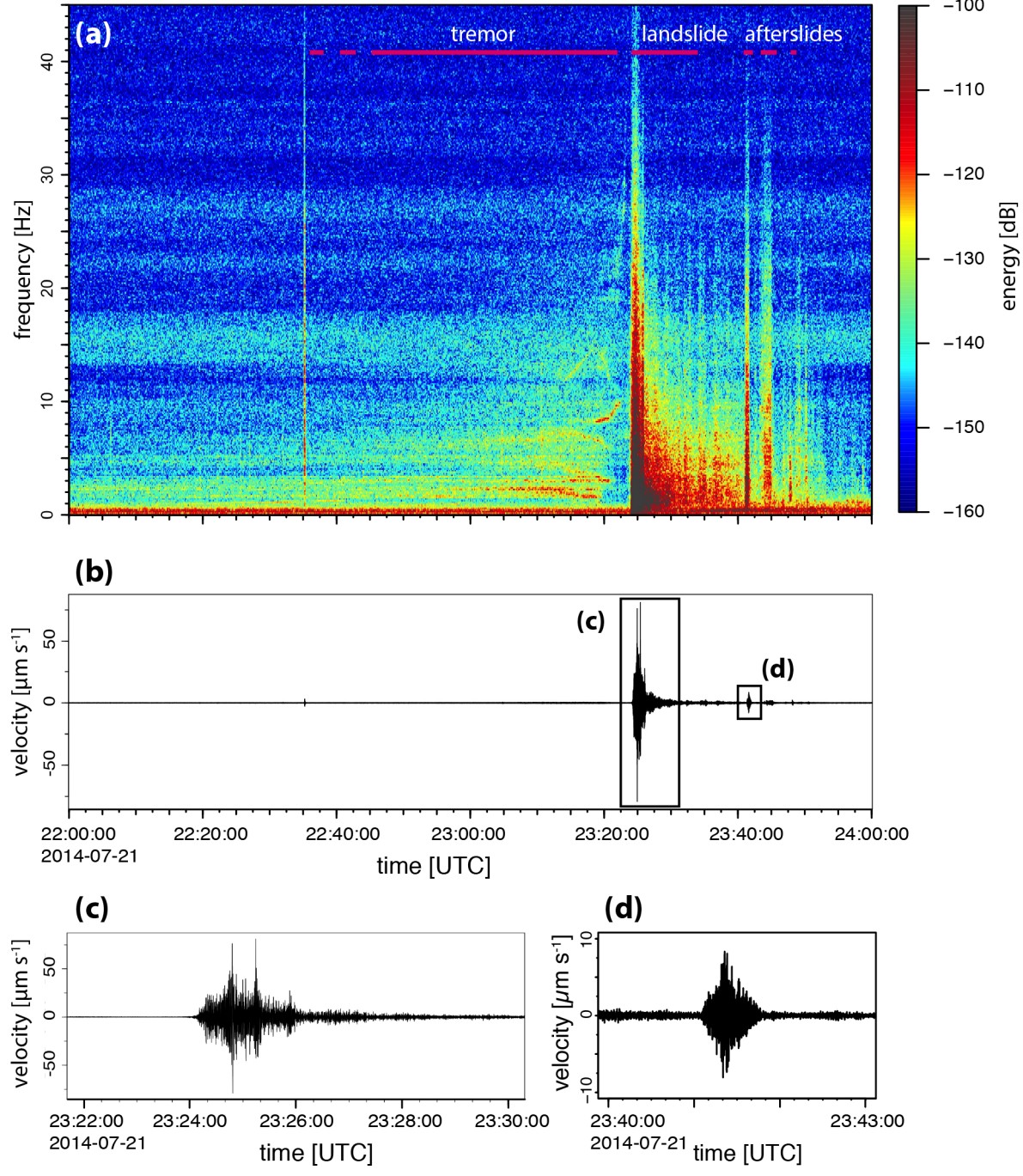

**Figure 4: (a) Spectrogram (unfiltered) and (b) waveform (filtered between 1–45 Hz) of the landslide, its precursory tremor and the afterslides at the Askja caldera, 21 July 2014. (c) Close-up of the landslide and (d) of one afterslide waveform. Station MOFO, East component.**

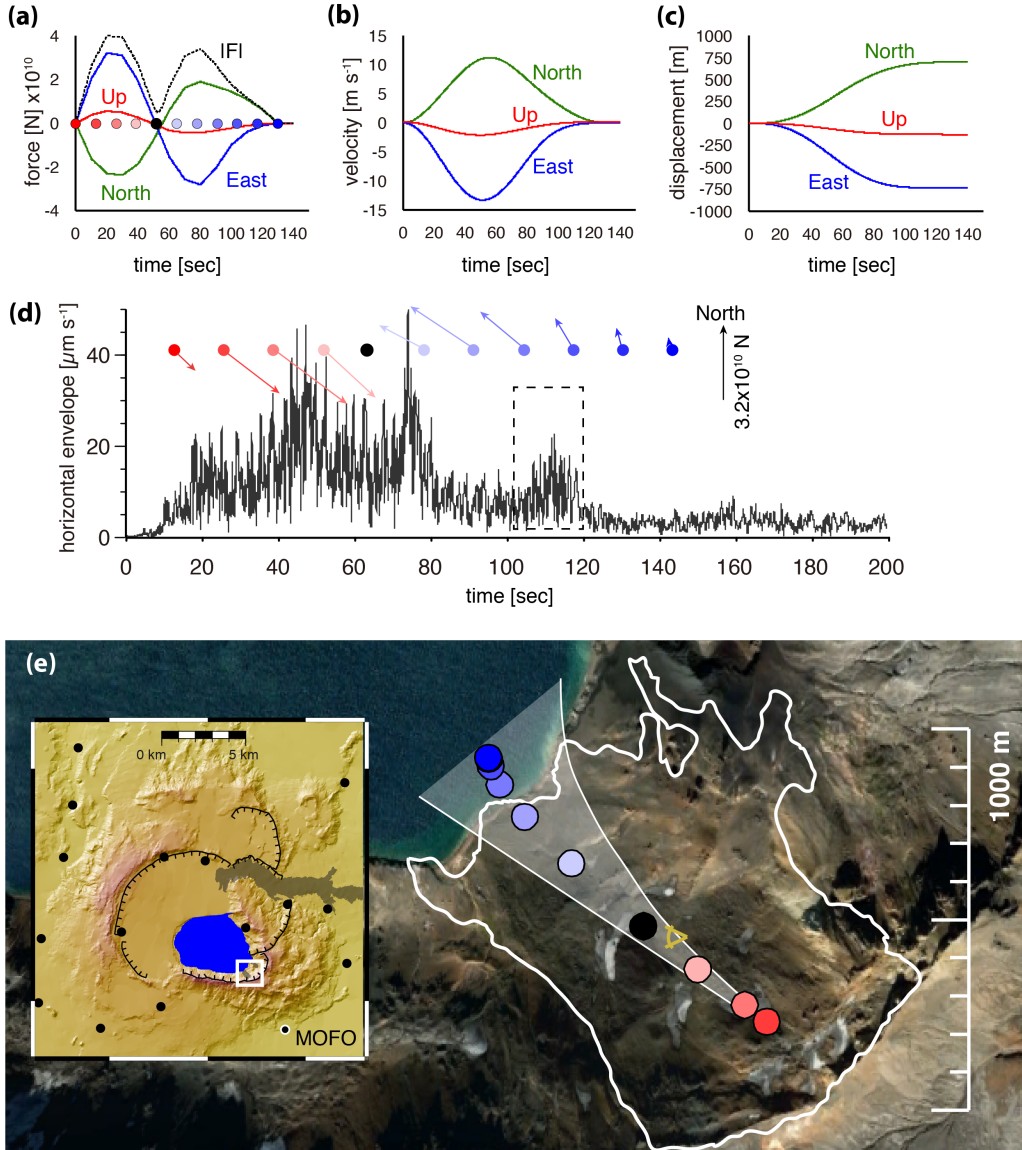

**Figure 5: Results of the landslide force history inversion. (a) Force-time evolution, (b) velocity-time and (c) displacement-time plots for the north, east and upwards component of the loading force. Red circles in a), d) and e) mark the acceleration phase and the black circle is the transition to the deceleration phase (blue circles) of the landslide motion. (d) Time evolution of the landslide acceleration and deceleration with horizontal force vectors (arrows) for each time step (North is up). Horizontal envelope function of station MOFO (see inset in (e) for location), computed with the root-mean-square amplitudes of the horizontal component waveforms, filtered between 1–45 Hz for comparison. The dashed box highlights the late-arriving seismic signals. The start of the x-axis is at 23:24:05 UTC. (e) Path of the landslide bulk mass from the landslide force history inversion of the seismic waveforms between 0.02–0.08 Hz. Shaded white area is the range of the inversion results with different frequencies of the band pass filter (0.02–0.05 Hz, 0.02–0.08 Hz, 0.04–0.08 Hz). The white line is the outline of the landslide source area plotted on top of a Google Earth image taken on 7 August 2012. The yellow eye looking W is the location of the photography presented in Fig. 3. The white square on the inset shows the location of the main image.**

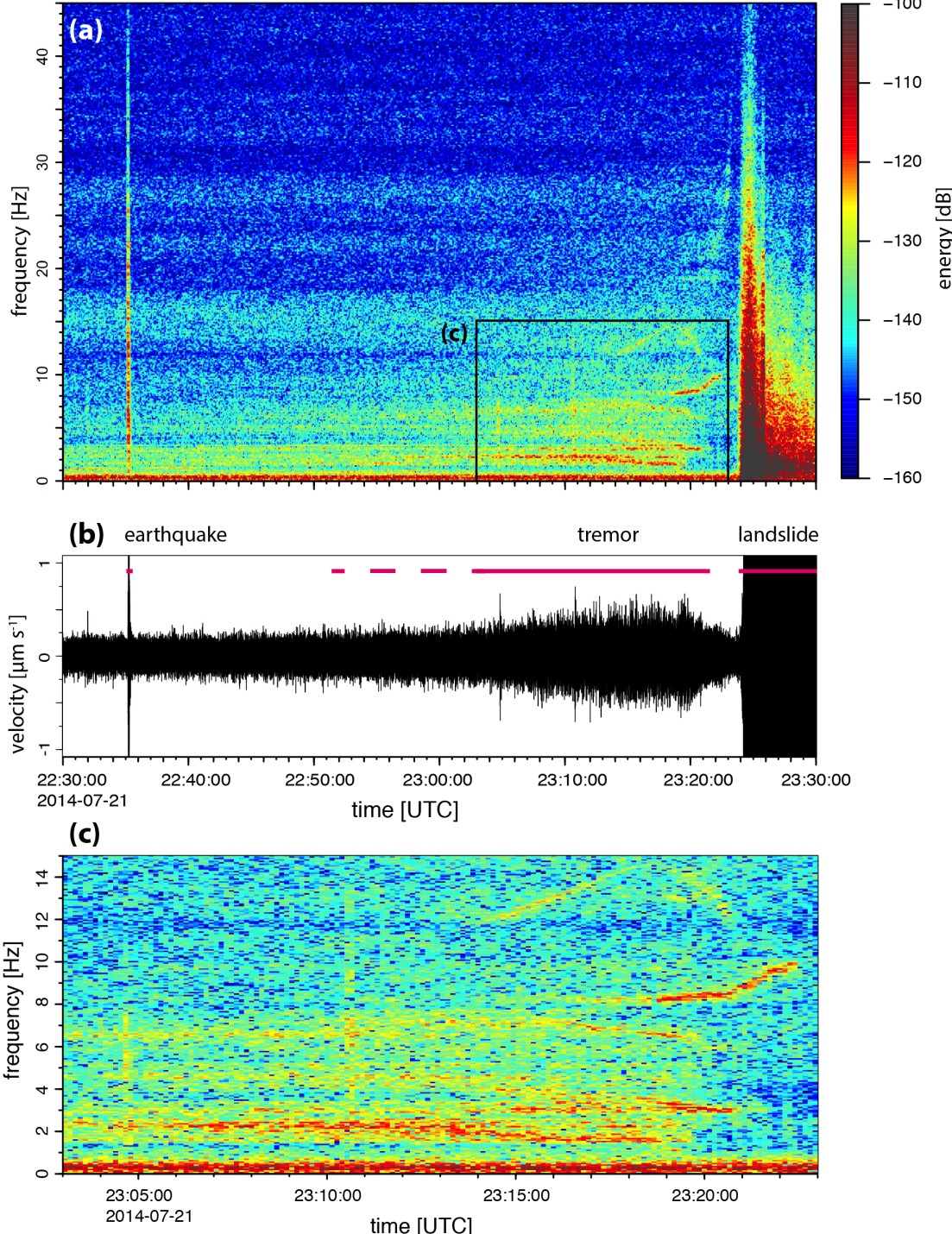

**Figure 6: (a)** Spectrogram (unfiltered) and **(b)** waveform (filtered between 1–45 Hz) of the tremor signal preceding the 21 July 2014 Askja landslide. **(c)** Close-up of the tremor signal with up- and down-gliding spectral lines. Station MOFO, East component.

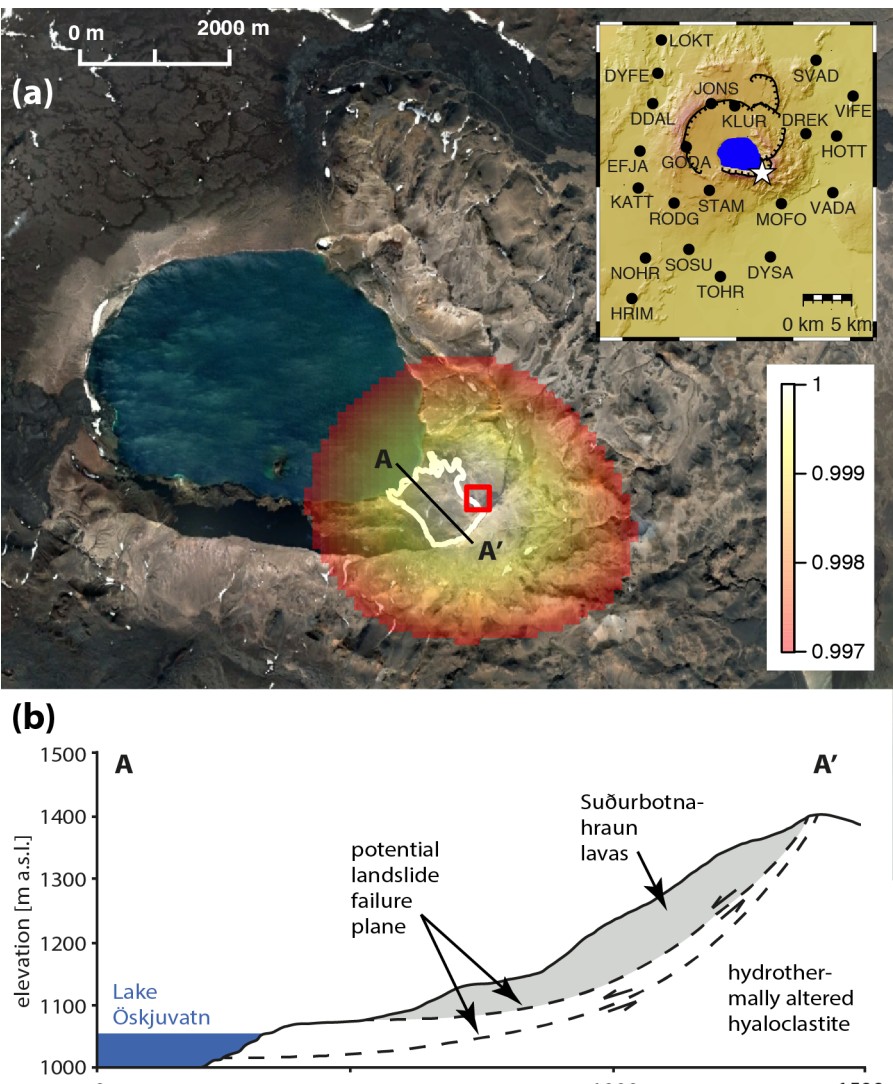

**Figure 7: (a) Example of the tremor localisation results using the tremor record of the stations shown in the inset between 23:00:00 and 23:01:00 UTC, filtered between 1.5–3 Hz and a seismic wave velocity of 2300m/s. The open red square is the best-fit location and the ellipse around it is the likelihood quantile from 0.997 (red) to 1 (translucent white). The white line is the outline of the landslide source area. The inset shows the locations of the seismic stations used in the localisation and the location of the landslide source area (white star). (b) Hypothetical cross section of the landslide showing the potential stick-slip tremor planes.**

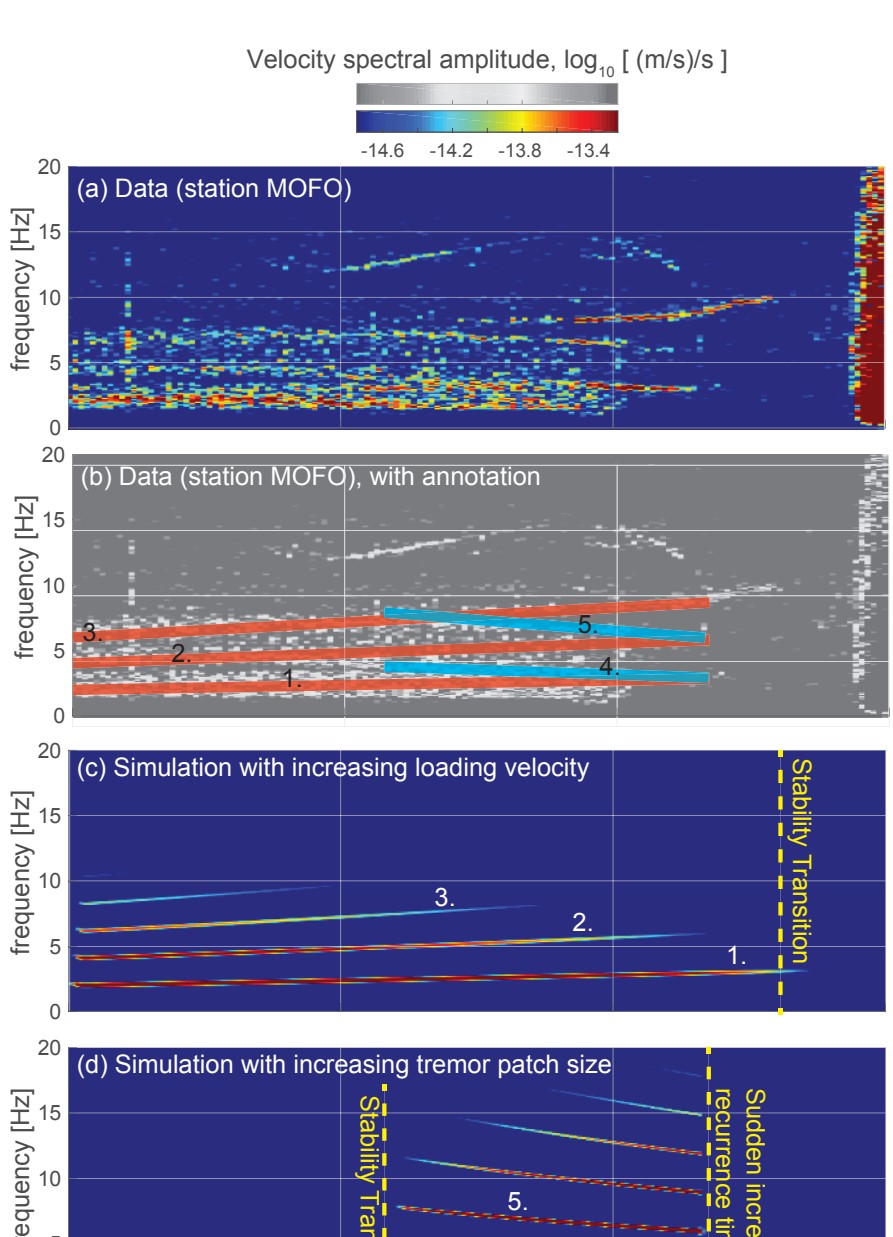

**Figure 8: Comparison between observed and simulated seismic tremor showing, (a) a data spectrogram from the station MOFO and (b) an annotated data spectrogram showing three up-gliding spectral bands (labelled 1, 2, 3) and two down-gliding spectral bands (labelled 4 and 5). The time of large-scale landslide motion is visible in the data at about 14 minutes time. (c) and (d) show numerical simulations with increasing loading velocity and increasing patch size, respectively. Simulation parameters are given in Table 1. The spectrograms in (a), (c), and (d) are created using the same colour scale and are therefore comparable. The start of the x-axis is at 23:10:00 UTC.**

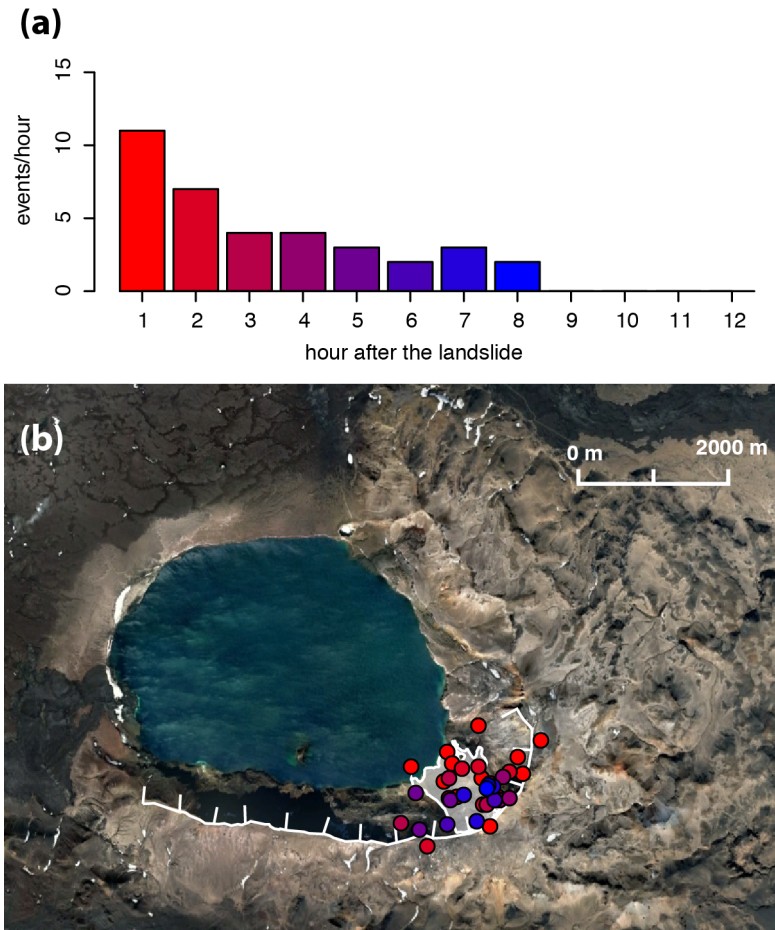

**Figure 9: (a) Numbers of small slope failures after the Askja landslide per hour. (b) Location of small slope failures after the Askja landslide, colour-coded by hour of occurrence as in (a). Only the best-fit locations are shown, the ellipses of the likelihood quantile are omitted for clarity (cf. Fig. 7a). The white area at the caldera ring fault is the landslide source area.**