# Peer review of "Dynamics of the Askja caldera July 2014 landslide, Iceland, from seismic signal analysis: precursor, motion and aftermath"

_Earth Surface Dynamics, 2017_

## Referee Comment (RC1) · J. Caplan-Auerbach (Referee) · 5 Jan 2018

This well-written paper describes a prolonged seismic signal associated with a large landslide in Askja caldera, and uses those data to describe the failure sequence. The authors use a variety of techniques to analyze the dynamics of the slide, including a precursory sequence and series of "afterslides". Overall, the paper provides an interesting description of this event, and much of the analysis is compelling. The identification of this signal is itself an important contribution, particularly given recent events such as the 2017 Nuugaatsiaq landslide/tsunami. However, I believe it requires more explanation and justification before it can be accepted for publication.

[Figure]

The landslide signal is preceded by a prolonged tremor sequence, which exhibits harmonics and apparent gliding. Immediately prior to the slide itself, the tremor stops and there is a period of quiescence. This is reminiscent of the signal recorded prior to eruptions at volcanoes such as Redoubt, and thus it is no surprise that the authors invoke a similar mechanism for the tremor signal (repeating, stick-slip events that occur at regular intervals). Furthermore, tremor-like signals have been recorded prior to other landslides, notably in Alaska.

That said, I found the analysis of the tremor and gliding to be somewhat weak. The authors use changes in the timing between stick-slip events as an explanation for the observed gliding. To explain the fact that both increases and decreases in frequency are observed, they propose that there are two discrete patches of slip that behave differently (one accelerates while the other slows). I don't deny that this is a possible model, but I don't find the justification that compelling. First, it isn't clear to me that the observed frequencies are actually harmonic. There is clear upgliding, but it is only a single frequency, and the downgliding is subtle and not obviously showing overtones. The modeling shown in Figure 8 does indeed confirm that repeated similar events can present as gliding, but they bear little resemblance to the gliding observed in the precursory tremor. Are the frequencies of observed and synthetic signals the same? Nowhere do the authors state which frequencies they believe represent up- and down-gliding, so it's difficult to tell. The synthetics have many more overtones than the observed signals. Can the authors explain this? The authors indicate that the model replicates the observed aseismic portion of the signal, but without knowing at what time the landslide would initiate in this model it's hard to tell if this model fits the data.

I understand the rationale for the two slip-patch model, but I'm not sure I buy it. That two patches could generate events similar enough to generate harmonics, and that both of those patches would experience regular acceleration or deceleration simultaneously seems unlikely. If we had evidence of strain on the order that is required for this

behavior, perhaps it would be plausible. But simply saying that this could be observed with "sufficiently high spatial resolution geodetic observations" is unsatisfying.

Another concern that I have with the modeling has to do with the force history analysis. The authors describe their modeling and describe their results, but we never see the results of the modeling (other than the location and history). The analysis describes the direction of motion, but this isn't presented; we only see the eastern component of velocity based off of the high frequency data. This needs to be much more thoroughly presented.

Note that a revised version of this paper should also cite Poli, 2017 (Poli, P. (2017), Creep and slip: Seismic precursors to the Nuugaatsiaq landslide (Greenland), Geophys. Res. Lett., 44, 8832–8836, doi:10.1002/2017GL075039) as it relates very directly to these processes. It might also be useful to read Kilburn and Petley 2003: (Kilburn, C. R., & Petley, D. N. (2003). Forecasting giant, catastrophic slope collapse: lessons from Vajont, Northern Italy. Geomorphology, 54(1), 21-32.) These are my broad concerns. Smaller issues within the text are enumerated below:

1. The abstract can be significantly shortened. There is a lot of detail within it that is unnecessary for an abstract: there's no need to include the motivation for the study, and much of the text can be cut out (e.g. change "The excellent seismic data quality and coverageof the stations of t the Askja network made it possible to jointly analyse. . ." to "we jointly analyzed. . .") 2. Page 2, line 4: "often" seems like a bit of an overstatement here. Tsunamigenic landslides on volcanoes have certainly occurred, but they are not common. 3. Page 2, line 25: I'm not sure there's any need to discuss iceberg tremor here; it's not relevant to the study. 4. Page 6, lines 24-25: While it's true that high frequencies attenuate more rapidly than low frequencies, I'm not sure that this is the reason for the shape of the spectrogram (it could also be a source mechanism). Perhaps the authors could comment on whether this shape is dependent on the distance to the seismometer? 5. Page 10, line 25: The authors describe 3.5 km as a long distance for seismic energy to be recorded. This actually strikes me as pretty close.

Perhaps the authors could comment on what distance they consider "close"? 6. Page 13, line 6: I recommend citing Norris, 1994 (Norris, R. D., 1994, Seismicity of rockfalls and avalanches at three Cascade Range volcanoes; implications for seismic detection of hazardous mass movements: Seismological Society of America Bulletin, v. 84, p. 1425–1939) as one of the earlier publications describing the appearance of seismic signals associated with landsliding. 7. Page 13, line 24: This line about the 38 seismometers within 30 km is repetitious. 8. Page 13, line 25: Unclear what the authors mean by "activated"? 9. Page 13, line 29: Is this just saying that there is an asperity on the failure plane?

---

## Referee Comment (RC2) · Anonymous Referee #2 · 9 Jan 2018

Dynamics of the Askja caldera July 2014 landslide, Iceland, from seismic signal analysis: precursor, motion and aftermath

This paper shows the precursory seismic signal of the Askja caldera landslide. It is well-written, and shows an interesting observation. Authors found that there were up-gliding and down-gliding signals in the seismic data before the landslide, and explained they were accelerating and decelerating stick-slip motion preceding the landslide. They reproduced this phenomenon by numerical modeling. The interpretations are interesting, but they are based on relatively strong assumptions. It is fine to use assumptions and make an interpretation, but in the discussion, the assumptions were treated almost

an accomplished fact. The proposition of landslide early warning is too optimistic after finding only one example of post-report. Please explain the mechanism more carefully or tone down the succeeding discussion.

[Major comments]

One of the assumptions I was not very convinced was that they treated the tremor as continuous stick-slip events with little intervals. It may be true, but the mechanisms of tremors are still debating and there are many other interpretations. Another brave assumption was that the frequency change was caused by the change of loading velocity. It was not easy for me to imagine the acceleration and deceleration of the velocity occur simultaneously at a single body (which is the assumption when you performed long-period inversion).

Many sentences in the discussion (section 6) were an interpretation based on the assumption, but they are discussed without considering assumptions (e.g. page 13 line 23-24, 26-28, page 14 line 17-18, page 14 lines 31-32). I think interpretations and observations should be classified.

In section 6.2, authors are discussed the possibility of potential landslide early warning system using seismic signal. The idea is interesting, but the way of writing seems to be too optimistic. At this moment we are drawing the target around the arrow. For example, the first sentence in page 15 says the precursory tremors should be dectable, but as authors may know, there are many landslides which do not generate precursory signal. For practical purpose, the success rate also should be investigated.

[Minor comments]

Page 4, section 2.1 and 2.2 The geological setting and seismicities are difficult to configure without map. Readers may not be familiar with the area. Since it seems they are used in the later interpretation, please add figures to explain them.

Page 6 lines 1-7 Please add a map to show those events.

**ESurfD**
Page 6 line 18 It is not clear for me what is cigar-like shape. Could you rephrase it? (same in page 13 line 3)

Page 7 section 3.2 Please show the force-time history and waveform fitting (possibly in appendix)

Page 7 line 13 Remove a big space between CMG-3Ts and instruments

Page 7 line 21 Inversion would give a force (mass times acceleration). How did you compute the mass?

Page 8 line 9 There are multiple lines and these three frequencies are not easy to identify. Could you zoom-in and add lines on the figure? How about the yellow curved line between 10-15 Hz?

Page 8 lines 13-17 Please show the spectrograms of other stations (e.g. DREK, GODA, HOTT, JONS, KLUR, MOFO, STAM, VADA) in the appendix.

Page 8 lines 18-19 I am not sure why this implies the surface wave. Other phases may give a large H/V.

Page 8 lines 22-27 Please show this amplitude ratio on a map.

Page 8 lines 28-32 Please show the time window used for this localization. The word of "migration" sounds confusing for me. In general, if you say migration for tremors, the source location will migrate as a function of time. If the location is fixed when you invert the location from the cross correlation time shift, I suggest to use gridsearch.

Page 10 lines 3-10 It would be helpful if you add a geological section of the landslide to understand this description.

Page 10 lines 8-9 Why does the higher energy transmit if the stick-slip motion happens at the sliding surface? That is not intuitive.

Page 10 line 12-17 This is quite strong assumption, and I was not convinced that it was

the only one possibility to explain this phenomenon. You wrote, "we deduce that the individual, repeating stick-slip events occurred very close together in time from the start of instability." On the other hand, "individual stick-slip events before the Askja landslide may not have been detectable kilometres away and that the events must occur already very close in time and transmit enough energy that they can be seen from a longer distance, likely as a continuous tremor signal." If you treat these tremors as summation of individual events, why the attenuation can be different? You wrote the individual events were not detectable farther away but tremor signal could transmit energy. That sounds contradictory for me.

Page 13 line 11-13 Please show the location of afterevents on a map.

Page 28 Figure 4 Why you use vertical component in Fig. 4 and EW component in Fig. 6?

Page 30 Figure 5 Please show the date of the photo (a) taken.

Page 31 Fig. 6 Why are there strong signals with frequency <1Hz after the bandpass filter between 1-45Hz?

Page 32 Figure 7 Please add a scalebar for the likelihood. Please add the definition of likelihood in the main text.

---

## Editor Comment (EC1) · K. Allstadt (Editor) · 16 Feb 2018

The authors presented a well-written and scientifically interesting case study with compelling findings. The two reviews were generally positive. While both reviewers did ask for more explanation and justification, the authors seem to addressed most of those concerns rigorously in their response to the interactive comments. I strongly encourage the authors to submit their revised manuscript. However, before doing so, I encourage the authors to ensure that they sufficiently address two of reviewer 2's comments that I did not feel were satisfactorily addressed in the response, as explained below:

Regarding page 7, line 21, reviewer 2 asks the authors to explain how they obtained

[Figure]

the mass from the force history, and pointed out that you must know the mass to get the trajectory or vice versa. In their response, the authors give an explanation of their methods, which involves an iterative process to match the trajectory to the observed runout distance. As the authors acknowledge in their original manuscript, the trajectory corresponds with that of the center of mass, not the overall runout distance. However, in Figure 5 of the initial submission, the authors place the first dot at the very top of the landslide, which is certainly not the initial location of the center of mass. Correcting for this would substantially shorten the trajectory and change all of the derived values. I also recommend the authors give some more explanation about how well they actually know the runout distance of the center of mass and how the uncertainty of that is reflected in the estimates they derived from the force history inversion. The location of the center of mass before and after the landslide can be challenging to estimate with just satellite imagery and field observations and a rigorous assessment requires some assumptions about the location of the failure plane. Is the uncertainty in the before and after location of the center of mass the source of the uncertainty ranges given on the estimated mass and other seismically-derived values in the original submission? No explanation is given but it is needed.

Additionally, regarding page 10, line 12-17, Reviewer 2 states "You wrote the individual events were not detectable farther away but tremor signal could transmit energy. That sounds contradictory for me." – I agree with reviewer 2 that it is contradictory to say that discrete repeating events are too weak to be recorded as individual events, but are strong enough to be recorded once they are closely spaced enough together to appear as tremor. Most if not all of the other studies that invoke repeating quakes as a source of gliding spectral lines have observed discrete events that become more frequent and grade into tremor. For stick-slip sliding, as the recurrence interval decreases, one might expect the subsequent earthquakes to become smaller rather than larger (slip-predictable behavior). The authors responded to this comment by saying that they cannot discern the individual slip events within the tremor, but that does not address the actual topic of concern of reviewer 2 which was that no discrete events were observed
BEFORE the tremor. Can the authors provide any more potential explanations for why no discrete events were ever observed if this tremor is, as they propose, generated by closely spaced repeating events that become more and more frequent?

I also have a few minor comments that were not already addressed by reviewers that should be addressed in the revision:

-Avoid overusing intensifiers like "very", "excellent", "exceptional" – they add little meaning. -pg3 L14-17 Where did the steam cloud come from if the event was a landslide? Mention hydrothermal depressurization up front, right now it's buried toward the end of the manuscript. -pg5 L20-24 Landslide triggering often depends on more than just a few days weather beforehand. It might be useful to put the weather data shown in Fig 2 in context of typical weather (i.e. was the entire period warmer than usual?). -pg5 L29 Is this seismic data openly available? If so, where? The authors can refer to a data and resources section or acknowledgements with details, but it's useful to state somewhere in the text. -pg6 L1 – CMM is not commonly known, it could use a brief explanation. -pg6 L12 – Ensure it is clear that the description here refers to the seismic signal from the catastrophic failure part of the landslide sequence, not the precursory tremor. -pg 7 L7-10 – It would be helpful the frequency ranges were also given in period in parentheses beside each range. -pg 7 L24 – The authors range is not actually within the other range, as stated. Perhaps instead the authors should say the ranges overlap? -pg8 L15-16 How was mean frequency computed? -pg 8 L23 Citation or more background needed on this method. -pg8 L30 – Using cross correlation of envelopes or time series waveforms? -pg10 L25-26 Citation needed -pg 11 L2-3 Can the authors provide any explanation for why the two different gliding bands seem to have different stopping points? -pg12 L26-28 This comparison does not seem useful as the two seem unrelated and likely have different mechanisms. Are the authors implying there is some link between the stick-slip events related to landslide motion and hydrothermal activity? -pg13 L23-24 Can the authors actually be sure the landslide didn't start moving before the observable tremor? The authors state earlier that they think there were unobservable discrete events leading up to this, meaning it must have started moving earlier than that. -pg14 L1 "accelerated" and "stable-sliding" seems like an oxymoron -pg14 L3 It is not clear why the authors cite other papers here for findings made regarding the present study.

---

## Author Comment (AC1) · 16 Feb 2018

**Authors' reply to comments of the reviewers**

Original title: Dynamics of the Askja caldera July 2014 landslide, Iceland, from seismic signal analysis: precursor, motion and aftermath

First submission: 30 November 2017

Manuscript No: esurf-2017-68

First reply to reviews: ??? 2018 submitted by A. Schöpa on behalf of the authors

We thank the reviewers for their constructive comments and suggestions. Please find in the following the reviewers' comments followed by our reply**.**

**Review 1 by J. Caplan-Auerbach**

This well-written paper describes a prolonged seismic signal associated with a large landslide in Askja caldera, and uses those data to describe the failure sequence. The authors use a variety of techniques to analyze the dynamics of the slide, including a precursory sequence and series of "afterslides". Overall, the paper provides an interesting description of this event, and much of the analysis is compelling. The identification of this signal is itself an important contribution, particularly given recent events such as the 2017 Nuugaatsiaq landslide/tsunami. However, I believe it requires more explanation and justification before it can be accepted for publication.

The landslide signal is preceded by a prolonged tremor sequence, which exhibits harmonics and apparent gliding. Immediately prior to the slide itself, the tremor stops and there is a period of quiescence. This is reminiscent of the signal recorded prior to eruptions at volcanoes such as Redoubt, and thus it is no surprise that the authors invoke a similar mechanism for the tremor signal (repeating, stick-slip events that occur at regular intervals). Furthermore, tremor-like signals have been recorded prior to other landslides, notably in Alaska. That said, I found the analysis of the tremor and gliding to be somewhat weak. The authors use changes in the timing between stick-slip events as an explanation for the observed gliding. To explain the fact that both increases and decreases in frequency are observed, they propose that there are two discrete patches of slip that behave differently (one accelerates while the other slows). I don't deny that this is a possible model, but I don't find the justification that compelling. First, it isn't clear to me that the observed frequencies are actually harmonic. There is clear upgliding, but it is only a single frequency, and the downgliding is subtle and not obviously showing overtones.

**Reply:** We added a zoom-in to Figure 6 and Figure 8 showing the up- and down-gliding spectral lines in more detail.

The modeling shown in Figure 8 does indeed confirm that repeated similar events can present as gliding, but they bear little resemblance to the gliding observed in the precursory tremor. Are the frequencies of observed and synthetic signals the same? Nowhere do the authors state which frequencies they believe represent up- and down-gliding, so it's difficult to tell.

**Reply:** We've modified our Figure 8 (and also our parameter choices, see Table 1) to highlight the similarities between the modelled and observed tremor. We believe that the up- and down-gliding tremor are now much more clearly shown.

The synthetics have many more overtones than the observed signals. Can the authors explain this?

**Reply:** It is true that some overtones that are not clearly visible in the data. The overtone labelled with a number two (Figure 8b and c) is less strongly observed than others, for example. We believe that the simplest explanation for this is that our simplified model of wave propagation fails to account for certain propagation phenomena that may diminish wave amplitudes. Wave propagation in the complicated, 3D, layered, attenuating media surrounding the Askja volcanic complex is far richer than we have attempted to capture.

The authors indicate that the model replicates the observed aseismic portion of the signal, but without knowing at what time the landslide would initiate in this model it's hard to tell if this model fits the data.

**Reply:** Our simulations have not attempted to couple the seismic tremor to the force-balance of large-scale land sliding motion. We find this to be a fascinating idea, although well beyond the scope of the present modelling efforts. We do believe, however, that our revised Figure 8 now more clearly emphasizes how the ways in which our model does fit several prominent aspects of the seismic data.

I understand the rationale for the two slip-patch model, but I'm not sure I buy it. That two patches could generate events similar enough to generate harmonics, and that both of those patches would experience regular acceleration or deceleration simultaneously seems unlikely. If we had evidence of strain on the order that is required for this behavior, perhaps it would be plausible. But simply saying that this could be observed with "sufficiently high spatial resolution geodetic observations" is unsatisfying.

**Reply:** This comment pushed us to rethink our explanation of down-gliding tremor. We agree that deceleration before the landslide is a physically unrealistic explanation. We find that we can equally well fit the data if the second tremor patch gradually expands in size. This explanation has an additional benefit as well. Since the seismic moment (and therefore the far-field seismic particle displacement amplitudes) is proportional to the patch area, this model is capable of explaining the increased tremor amplitudes before the quiescent period. We describe these improvements in greater detail in Section 4.

Another concern that I have with the modeling has to do with the force history analysis. The authors describe their modeling and describe their results, but we never see the results of the modeling (other than the location and history). The analysis describes the direction of motion, but this isn't presented; we only see the eastern component of velocity based off of the high frequency data. This needs to be much more thoroughly presented.

**Reply:** We added the results of the modelling, namely the force-time history, the velocity vs. time, the displacement vs. time, and the directions of motion vs. time plots to Figure 5. We also now present the waveform fits of the recorded and synthetic waveforms in Figure S3 of the supplement.

Note that a revised version of this paper should also cite Poli, 2017 (Poli, P. (2017), Creep and slip: Seismic precursors to the Nuugaatsiaq landslide (Greenland), Geophys. Res. Lett., 44, 8832–8836, doi:10.1002/2017GL075039) as it relates very directly to these processes.

**Reply:** We now cite Poli's work.

It might also be useful to read Kilburn and Petley 2003: (Kilburn, C. R., & Petley, D. N. (2003). Forecasting giant, catastrophic slope collapse: lessons from Vajont, Northern Italy. Geomorphology, 54(1), 21-32.)

**Reply:** We also refer to Kilburn and Petley (2003) in the revised version.

These are my broad concerns. Smaller issues within the text are enumerated below:

1. The abstract can be significantly shortened. There is a lot of detail within it that is unnecessary for an abstract: there's no need to include the motivation for the study, and much of the text can be cut out (e.g. change "The excellent seismic data quality and coverageof the stations of t the Askja network made it possible to jointly analyse. . ." to "we jointly analyzed. . .")

**Reply:** We shortened and condensed the abstract.

2. Page 2, line 4: "often" seems like a bit of an overstatement here. Tsunamigenic landslides on volcanoes have certainly occurred, but they are not common.

**Reply:** We deleted the word "often".

3. Page 2, line 25: I'm not sure there's any need to discuss iceberg tremor here; it's not relevant to the study.

**Reply:** We deleted the sentence about iceberg tremor.

4. Page 6, lines 24-25: While it's true that high frequencies attenuate more rapidly than low frequencies, I'm not sure that this is the reason for the shape of the spectrogram (it could also be a source mechanism). Perhaps the authors could comment on whether this shape is dependent on the distance to the seismometer?

**Reply:** We changed the wording of the sentence and now also refer to source effects.

5. Page 10, line 25: The authors describe 3.5 km as a long distance for seismic energy to be recorded. This actually strikes me as pretty close. Perhaps the authors could comment on what distance they consider "close"?

**Reply:** Seismic studies we are aware of (Amitrano et al., 2005; Zeckra et al., 2015, Yamada et al., 2016) that report on cracking before slope collapses had instruments located less than a kilometre away from the collapse area. Relative to this, we considered >3.5 km as long. We clarified this in the manuscript.

6. Page 13, line 6: I recommend citing Norris, 1994 (Norris, R. D., 1994, Seismicity of rockfalls and avalanches at three Cascade Range volcanoes; implications for seismic detection of hazardous mass movements: Seismological Society of America Bulletin, v. 84, p. 1425–1939) as one of the earlier publications describing the appearance of seismic signals associated with landsliding.

**Reply:** We added this reference.

7. Page 13, line 24: This line about the 38 seismometers within 30 km is repetitious.

**Reply:** We rephrased this sentence and deleted the repetition.

8. Page 13, line 25: Unclear what the authors mean by "activated"?

**Reply:** We meant excited and deleted the word "activated".

9. Page 13, line 29: Is this just saying that there is an asperity on the failure plane?

**Reply:** Correct. We now use the term asperity.

**Review 2 by Anonymous Referee #2**

This paper shows the precursory seismic signal of the Askja caldera landslide. It is well-written, and shows an interesting observation. Authors found that there were up- gliding and down-gliding signals in the seismic data before the landslide, and explained they were accelerating and decelerating stick-slip motion preceding the landslide. They reproduced this phenomenon by numerical modeling. The interpretations are interesting, but they are based on relatively strong assumptions. It is fine to use assumptions and make an interpretation, but in the discussion, the assumptions were treated almost an accomplished fact. The proposition of landslide early warning is too optimistic after finding only one example of post-report. Please explain the mechanism more carefully or tone down the succeeding discussion.

[Major comments]

One of the assumptions I was not very convinced was that they treated the tremor as continuous stick-slip events with little intervals. It may be true, but the mechanisms of tremors are still debating and there are many other interpretations.

**Reply:** This comment has motivated us to greatly rethink and rewrite Section 4 of our manuscript. The principal change has been to provide a clarified justification of our reasoning concerning the mechanical origin of the observed seismic tremor. As a general comment, we believe that the field of tremor source process modelling has greatly advanced in the fifteen years since the discovery of subduction zone episodic tremor and slip (i.e., Rogers and Dragert, 2003, Science). This discovery lead to a dramatic increase in the number of mechanical modelling studies of various tremor source processes. We have provided a streamlined review of this (and other) literature with emphasis on how the tremor observed at Askja may be generated.

Another brave assumption was that the frequency change was caused by the change of loading velocity. It was not easy for me to imagine the acceleration and deceleration of the velocity occur simultaneously at a single body (which is the assumption when you performed long-period inversion).

**Reply:** Prompted by this comment, as well as a similar comment from Referee #1, we have greatly rethought our explanation of the tremor frequency change. In particular, we no longer favour the explanation of local deceleration prior to the landslide. As we wrote to Referee #1, "We agree that deceleration before the landslide is a physically unrealistic explanation. We find that we can equally well fit the data if the second tremor patch gradually expands in size. This explanation has an additional benefit as well. Since the seismic moment (and therefore the far-field seismic particle displacement amplitudes) is proportional to the patch area, this model is capable of explaining the increased tremor amplitudes before the quiescent period. We describe these improvements in greater detail in Section 4."

Many sentences in the discussion (section 6) were an interpretation based on the assumption, but they are discussed without considering assumptions (e.g. page 13 line 23-24, 26-28, page 14 line 17-18, page 14 lines 31-32). I think interpretations and observations should be classified.

**Reply:** We changed the wording of this section and pointed out the assumptions and interpretations more clearly.

In section 6.2, authors are discussed the possibility of potential landslide early warning system using seismic signal. The idea is interesting, but the way of writing seems to be too optimistic. At this moment we are drawing the target around the arrow. For example, the first sentence in

page 15 says the precursory tremors should be dectable, but as authors may know, there are many landslides which do not generate precursory signal. For practical purpose, the success rate also should be investigated.

**Reply:** We have modified our discussion of the potential to use tremor as an early warning system. In particular, we have changed the tone of our argument to be more modest. We simply wish to imply that the feasibility of such as system deserves further study.

[Minor comments]

Page 4, section 2.1 and 2.2 The geological setting and seismicities are difficult to configure without map. Readers may not be familiar with the area. Since it seems they are used in the later interpretation, please add figures to explain them.

**Reply:** The inset of Figure 1 already shows the location names mentioned in section 2.1 but we added more references to this figure in the text. We added the location names mentioned in section 2.2 to Figure 1.

Page 6 lines 1-7 Please add a map to show those events.

**Reply:** We added two maps to the supplement, one showing the earthquakes in the month before the landslide and one with the earthquakes occurring one month after the landslide (Figs. S1 and S2).

Page 6 line 18 It is not clear for me what is cigar-like shape. Could you rephrase it? (same in page 13 line 3)

**Reply:** To show the shape of the waveforms in detail, we added a zoom-in of the landslide signal and of one afterslide signal to Figure 4. We now use the term spindle-shaped.

Page 7 section 3.2 Please show the force-time history and waveform fitting (possibly in appendix)

**Reply:** We added the force-time history to Figure 5 and present the waveform fits in Figure S3 of the supplement.

Page 7 line 13 Remove a big space between CMG-3Ts and instruments

**Reply:** We removed the space.

Page 7 line 21 Inversion would give a force (mass times acceleration). How did you compute the mass?

**Reply:** We assumed a constant value for the landslide mass ($m$) and calculated the acceleration time-series by dividing the force results by the mass (a = F / m). Then, the displacement ($d$, the trajectory of the sliding mass) can be computed from the second integration of the acceleration. Thus, we can estimate the mass ($m$) from the resulted force time history that is also able to explain the run-out distance (~ 1000 m) and the run-out trajectories of the event inferred from remote sensing images and field observations.
Chao et al. (2016, Scientific Reports) present the estimated mass of 10 large-scale landslides and more information related to mass computations can be found there.

Page 8 line 9 There are multiple lines and these three frequencies are not easy to identify. Could you zoom-in and add lines on the figure? How about the yellow curved line between 10-15 Hz?

**Reply:** We added a zoom-in to the figure showing the up- and down-gliding spectral lines in more detail (Figs. 6 and 8).

Page 8 lines 13-17 Please show the spectrograms of other stations (e.g. DREK, GODA, HOTT, JONS, KLUR, MOFO, STAM, VADA) in the appendix.
**Reply:** We now present the stacked and single-station spectrograms in the supplement (Figs. S4 and S5).

Page 8 lines 18-19 I am not sure why this implies the surface wave. Other phases may give a large H/V.
**Reply:** We acknowledge that a large H/V ratio may be produced by several phases and changed the wording of this sentence.

Page 8 lines 22-27 Please show this amplitude ratio on a map.
**Reply:** We added a map showing the amplitude ratios to the supplement (Fig. S6).

Page 8 lines 28-32 Please show the time window used for this localization.
**Reply:** We added a sentence to the manuscript saying that "We used a frequency range of 1.5-3 Hz as this frequency band shows the highest tremor energy and time windows of 1 min starting at 22:54:00 UTC."

The word of "migration" sounds confusing for me. In general, if you say migration for tremors, the source location will migrate as a function of time. If the location is fixed when you invert the location from the cross correlation time shift, I suggest to use gridsearch.
**Reply:** We are sorry for the confusion. The term "migration" is taken from the original paper presenting the location method saying that "the migration of these observed time delays, that is, the conversion from time to distance, can be used to retrieve the origin of an event" (Burtin et al. 2014, Earth Surface Dynamics). To avoid further confusion, we changed the term in the manuscript.

Page 10 lines 3-10 It would be helpful if you add a geological section of the landslide to understand this description.
**Reply:** We changed this section substantially and clarified the language. We added a speculative cross section to Figure 7 showing the hypothetical failure planes.

Page 10 lines 8-9 Why does the higher energy transmit if the stick-slip motion happens at the sliding surface? That is not intuitive.
**Reply:** If the landslide mass is more damaged than its foundation, we expect it to act as a low velocity waveguide and hence transmit less energy into the solid Earth. We've modified the language on this point.

Page 10 line 12-17 This is quite strong assumption, and I was not convinced that it was the only one possibility to explain this phenomenon. You wrote, "we deduce that the individual, repeating stick-slip events occurred very close together in time from the start of instability." On the other hand, "individual stick-slip events before the Askja landslide may not have been detectable kilometres away and that the events must occur already very close in time and transmit enough energy that they can be seen from a longer distance, likely as a continuous tremor signal." If you treat these tremors as summation of individual events, why the attenuation can be different? You wrote the individual events were not detectable farther away but tremor signal could transmit energy. That sounds contradictory for me.
**Reply:** These sentences were written in a confusing way. We meant to say that we can distinguish the presence of the seismic tremor versus its absence. The tremor itself, as we

have shown in numerical simulations, may be explained as due to many repeating velocity pulses that are blurred together due to a combination of effects. Hence we cannot discern individual slip events within the seismic tremor. We've heavily modified the writing surrounding these sentences.

Page 13 line 11-13 Please show the location of afterevents on a map.
**Reply:** We added a map with the afterslides to Figure 9 showing the best fit locations of the small slope failures.

Page 28 Figure 4 Why you use vertical component in Fig. 4 and EW component in Fig. 6?
**Reply:** We changed Figure 4 and present the EW component there as well. This component shows the highest amplitudes for the landslide and the tremor.

Page 30 Figure 5 Please show the date of the photo (a) taken.
**Reply:** We added to the figure caption that the background image from Google Earth was taken on 7 August 2012.

Page 31 Fig. 6 Why are there strong signals with frequency <1Hz after the bandpass filter between 1-45Hz?
**Reply:** We apologise that the figure caption was misleading. The spectrogram is not filtered whereas the waveform is. We clarified this in the figure caption.

Page 32 Figure 7 Please add a scalebar for the likelihood. Please add the definition of likelihood in the main text.
**Reply:** We added a scale for the likelihood in the figure and now mention in the text that "we used the location procedure of Burtin et al. (2013) to locate the tremor signal on a DEM grid. This statistical approach assigns a probability to each grid point that it is the source of the signal based on cross-correlation of the waveforms at different stations. The resulting probability density function is normalised to its maximum value giving this grid point a likelihood of 1 to be the source location of the signal (Burtin et al., 2014)."

---

## Author Comment (AC2) · 15 Mar 2018

**Authors' reply to comments of the editor**

Original title:              Dynamics of the Askja caldera July 2014 landslide,
                             Iceland, from seismic signal analysis: precursor, motion
                             and aftermath
First submission:            30 November 2017
Manuscript No:               esurf-2017-68
First reply to reviews:      16 February 2018 submitted by A. Schöpa on behalf of
                             the authors

We thank the editor for her constructive comments and suggestions. Please find in
the following the editor's comments followed by our reply**.**

**Comments by K. Allstadt (Editor)**

The authors presented a well-written and scientifically interesting case study with
compelling findings. The two reviews were generally positive. While both reviewers did
ask for more explanation and justification, the authors seem to addressed most of those
concerns rigorously in their response to the interactive comments. I strongly encourage
the authors to submit their revised manuscript. However, before doing so, I encourage the
authors to ensure that they sufficiently address two of reviewer 2's comments that I did
not feel were satisfactorily addressed in the response, as explained below:
Regarding page 7, line 21, reviewer 2 asks the authors to explain how they obtained the
mass from the force history, and pointed out that you must know the mass to get the
trajectory or vice versa. In their response, the authors give an explanation of their
methods, which involves an iterative process to match the trajectory to the observed
runout distance. As the authors acknowledge in their original manuscript, the trajectory
corresponds with that of the center of mass, not the overall runout distance. However, in
Figure 5 of the initial submission, the authors place the first dot at the very top of the
landslide, which is certainly not the initial location of the center of mass. Correcting for
this would substantially shorten the trajectory and change all of the derived values. I also
recommend the authors give some more explanation about how well they actually know
the runout distance of the center of mass and how the uncertainty of that is reflected in
the estimates they derived from the force history inversion. The location of the center of
mass before and after the landslide can be challenging to estimate with just satellite
imagery and field observations and a rigorous assessment requires some assumptions
about the location of the failure plane. Is the uncertainty in the before and after location
of the center of mass the source of the uncertainty ranges given on the estimated mass
and other seismically-derived values in the original submission? No explanation is given
but it is needed.
**Reply:** The editor's point is well taken. In this study, we performed an inversion of the
long-period seismic signals between 0.02-0.08 Hz based on a model where the landslide
mass is treated as a single block. The spatial scale of the landslide event is small enough
compared to the wavelength of above filtered seismic waves to satisfy the block model
approximation (i.e. seismic point source). Assuming a constant value of landslide mass
over time, we can estimate the acceleration time-series by dividing the inverted forces by

the landslide mass. Then, we put the initial position of the block mass at the top of the landslide area and find the mass by ensuring that the block-mass trajectory inferred from acceleration time-series matches the run-out path from satellite images and field observations. Here, the landslide mass for the grid-search scheme ranged from $1 \times 10^{10}$ kg to $2 \times 10^{11}$ kg.

Uncertainties in the inversion result mainly rely on the data quality of observed seismograms. Chao et al. (2016, 2017) demonstrated that only few stations with good signal-to-noise ratios (SNR) are sufficient to produce a reliable inversion result (i.e. waveform fitness value is larger than 0.75). In fact, the SNR of the waveforms depends on the frequency range of the band-pass filter. In order to test the sensitivity of the waveform inversion to the chosen frequency band, we used three frequency ranges of 0.02-0.05 Hz, 0.02-0.08 Hz and 0.04-0.08 Hz. The most likely run-out path of the landslide event is obtained from averaging over all the inversion results taking the standard deviation into account (see Fig. 5e). This is reflected in the range of values giving for the horizontal displacement of 1260±250 m, the vertical displacement of 430±300m and the landslide mass of 7-16$\times 10^{10}$ kg. We have revised the manuscript to clarify the above statements.

Additionally, regarding page 10, line 12-17, Reviewer 2 states "You wrote the individual events were not detectable farther away but tremor signal could transmit energy. That sounds contradictory for me." – I agree with reviewer 2 that *it is contradictory to say that discrete repeating events are too weak to be recorded as individual events*, but are strong enough to be recorded once they are closely spaced enough together to appear as tremor. Most if not all of the other studies that invoke repeating quakes as a source of gliding spectral lines have observed discrete events that become more frequent and grade into tremor. For stick-slip sliding, as the recurrence interval decreases, one might expect the subsequent earthquakes to become smaller rather than larger (slip- predictable behavior). The authors responded to this comment by saying that they cannot discern the individual slip events within the tremor, but that does not address the actual topic of concern of reviewer 2, which was that no discrete events were observed BEFORE the tremor. *Can the authors provide any more potential explanations for why no discrete events were ever observed if this tremor is, as they propose, generated by closely spaced repeating events that become more and more frequent?*

**Reply:** Our original presentation of this idea was in need of improvement. We have now included supplemental Figure S7, which shows a sample seismogram, and its Fourier transform. This seismogram shows that individual events are in fact discernable, but that they are rather jumbled in appearance and difficult to interpret. The Fourier velocity spectrum, in contrast, shows clear spectral peaks whose interpretation we give in the main text.

There is a one-to-one correlation between the spectral peaks due to a repeating signal, and the repeating signal itself. It is not possible to have one without the other. However, because we infer the presence of multiple superposed signals, the interpretation of the time domain signal can be difficult. It is possible, for example, that a positive velocity pulse from one tremor patch and a negative velocity pulse from another, distinct tremor patch arrives at the same time and cause destructive interference. Other complexities are also possible. We have revised the main text to emphasize that while individual events

are possible, we find that the time averaging afforded by a spectrogram facilitates interpretation of the observed seismic signal.

Several other studies of seismic tremor due to repeating stick-slip have also noted this time domain/frequency domain phenomenology. Dmitrieva et al. (2013) specifically noted that no individual events were present in their study on Mt. Redoubt Volcano. In studies of the Whillans Ice Plain, Lipovsky and Dunham (2016) and Winberry et al. (2013) observed some locations that had clear individual events and other locations that had tremor signals where individual events were blurred together. We have now also noted this in the text.

I also have a few minor comments that were not already addressed by reviewers that should be addressed in the revision:

-Avoid overusing intensifiers like "very", "excellent", "exceptional" – they add little meaning.

**Reply:** We went through the text and deleted those adjectives where appropriate.

-pg3 L14-17 Where did the steam cloud come from if the event was a landslide? Mention hydrothermal depressurization up front, right now it's buried toward the end of the manuscript.

**Reply:** We rephrased this sentence and now refer to the depressurization of the hydrothermal system. We now also mention that the cloud could have also contained a considerable amount of dust from the landslide.

-pg5 L20-24 Landslide triggering often depends on more than just a few days weather beforehand. It might be useful to put the weather data shown in Fig 2 in context of typical weather (i.e. was the entire period warmer than usual?).

**Reply:** We computed a mean July temperature for the year 2014 of 8.1°C for weather station Kárahnjúkar, and now mention in the text that "the area around the Askja central volcano experienced a period of warm weather in July 2014 with a mean monthly temperature of 8°C, 2 degrees higher than the long-term average of the mean July temperature."

-pg5 L29 Is this seismic data openly available? If so, where? The authors can refer to a data and resources section or acknowledgements with details, but it's useful to state somewhere in the text.

**Reply:** We added a data availability section saying that "the seismic dataset is available upon request from Prof. Robert White, Bullard Laboratories, University of Cambridge, Cambridge CB3 0EZ, United Kingdom."

-pg6 L1 – CMM is not commonly known, it could use a brief explanation.

**Reply:** We added a short description of this earthquake location algorithm saying that "this method combines seismic imaging and travel time inversion to determine the locations and times of earthquakes from seismic data recorded continuously on a sparse local seismometer array. Data inversion is done as a 3D subsurface grid search over the data and network of trial locations for likely locations and origin times of seismic events

(Drew et al., 2013).”

-pg6 L12 – Ensure it is clear that the description here refers to the seismic signal from the catastrophic failure part of the landslide sequence, not the precursory tremor.
**Reply:** We changed this sentence and now refer to the "high-amplitude short-period signals generated by the catastrophic failure part of the Askja landslide sequence".

-pg 7 L7-10 – It would be helpful the frequency ranges were also given in period in parentheses beside each range.
**Reply:** We added the period rages to the frequencies.

-pg 7 L24 – The authors range is not actually within the other range, as stated. Perhaps instead the authors should say the ranges overlap?
**Reply:** We changed this statement and now speak of overlapping ranges.

-pg8 L15-16 How was mean frequency computed?
**Reply:** We clarified this procedure in the manuscript by saying "first, we stacked the spectrograms of the eight closest stations, which where computed with the same specifications like time window length and fraction of window overlap, etc., by adding their energy values per frequency and time step. Then, we divided these sums by the number of stations to obtain the mean energies for the frequencies and time period of interest. The result shows that the gliding spectral lines of the tremor are clearly visible as sharp bands of higher energy values and did not become blurred in the stacked spectrogram (Fig. S5)."

-pg 8 L23 Citation or more background needed on this method.
**Reply:** We added more detail to this method and now state that "we computed the ratios of the mean envelope amplitudes of 1 minute of the tremor to 3 minutes of background seismic noise for all stations of the network. First, we removed the instrument response, the mean and the trend, and band-pass filtered the signals between 1-45 Hz. Then, we computed the envelopes for 1 minute of the tremor starting at 23:17:00 UTC, 21 July 2014, and for 3 minutes of background seismic noise starting at 00:10:00 UTC of the same day for the E components. Next, we calculated the mean amplitudes of the envelopes for these two time windows and determined their ratio."

-pg8 L30 – Using cross correlation of envelopes or time series waveforms?
**Reply:** We used signal envelopes for the cross-correlation. We clarified this in the manuscript.

-pg10 L25-26 Citation needed.
**Reply:** We changed this section completely in the new version of the manuscript.

-pg 11 L2-3 Can the authors provide any explanation for why the two different gliding bands seem to have different stopping points?
**Reply:** In the new version of the manuscript, we favour the explanation that the up- and down-gliding spectral lines of the tremor are caused by an accelerating (visible in the upgliding) and a growing (down-gliding) stick-slip patch. As these two patches seem to be controlled by different mechanisms and move independently, we envisage that the patches transition into a state of seismically non-detectable movement due to different reasons, which might happen at different times. In the manuscript, we explain the disappearance of the tremor as follows:

"These simulations additionally predict the disappearance of the tremor signal shortly before the landslide. We suggest that two different mechanisms are responsible for this behaviour in our case. First, the patch that experiences accelerated loading eventually crosses the stability threshold and begins to start sliding stably (R < Rc in Eq. 1). This behaviour is consistent with the theoretical prediction of a transition from stick-slip to stable sliding at high loading rates (Rice et al., 2001; Gomberg et al., 2011). In the simulations, this can be traced by the up-gliding spectral lines whose energy contents decrease with time until they fade into the background at 13 minutes (Fig. 8c). Second, the patch with growing area experiences a commensurate increase in recurrence time (recurrence time and patch size are proportional, see Eq. 2); eventually the recurrence time becomes so large that a quiescent period ensues. This can be seen in the simulations of the down-gliding spectral lines that disappear at 12 minutes (Fig. 8d). "

-pg12 L26-28 This comparison does not seem useful as the two seem unrelated and likely have different mechanisms. Are the authors implying there is some link between the stick-slip events related to landslide motion and hydrothermal activity?
**Reply:** We concur with the editor that the stick-slip events and the earthquakes of the hydrothermal system likely result from different physical mechanisms. Therefore, we changed this paragraph.

-pg13 L23-24 Can the authors actually be sure the landslide didn't start moving before the observable tremor? The authors state earlier that they think there were unobservable discrete events leading up to this, meaning it must have started moving earlier than that.
**Reply:** We agree that the landslide might have moved before the observable tremor and clarified this in the manuscript.

-pg14 L1 "accelerated" and "stable-sliding" seems like an oxymoron
**Reply:** We changed the wording of this sentence. It now reads "Through acceleration and growth of the sliding planes, the stick-slip sliding transitioned into an aseismic, stable sliding period".

-pg14 L3 It is not clear why the authors cite other papers here for findings made regarding the present study.
**Reply:** The concept of distinguishing the phases of landslide motion based on seismic signals was introduced by Hibert et al. (2014) and Chao et al. (2016). We changed the sentence in the manuscript to "based on combined inspection of the high- and low-frequency signals generated by the Askja landslide, we distinguish three phases of landslide motion, initiation, propagation and termination as proposed by Hibert et al. (2014) and Chao et al. (2016). "

---

## Author Response (AR1)

[revised manuscript text omitted]

Furthermore, Lipovsky and Dunham (2015) analysed seismic tremor due to hydraulic resonance and found that the resonant frequencies of a hydraulic fracture are expected to be unevenly spaced following $f\_n/f\_1 = n^{(3/2)}$. In contrast, Lipovsky and Dunham (2016) showed that a simple application of the Fourier transform to a repeating sequence of slip pulses results in a frequency pattern of $f\_n/f\_1 = n$. For a fundamental tone of $f\_1 = 2.5$ Hz as observed for the Askja landslide tremor, we would expect its first harmonic at 7.1 Hz for a resonating fracture or at 5 Hz for a stick-slip source. Observations of the Askja landslide tremor show that the spectral peaks are relatively evenly spaced with $f\_1 = 2.5$ Hz, $f\_2 = 5.0$ Hz, and $f\_3 = 7.5$ Hz giving rise to the pattern that $f\_n/f\_1 = n$. This provides observational evidence for a stick-slip tremor mechanism and against a hydraulic tremor source.

Several additional lines of reasoning support the interpretation of tremor as being due to small, repeating earthquakes along the landslide failure plane. Small, repeating earthquakes 
[revised manuscript text omitted]

---

## Author Response (AR2)

**Authors' reply to comments of the reviewers and the editor**

| | |
|---|---|
| Original title: | Dynamics of the Askja caldera July 2014 landslide, Iceland, from seismic signal analysis: precursor, motion and aftermath |
| First submission: | 30 November 2017 |
| Manuscript No: | esurf-2017-68 |
| First reply to reviews: | 16 February 2018 submitted by A. Schöpa on behalf of the authors |

We thank the reviewers and the editor for their constructive comments and suggestions. Please find in the following the reviewers' and editor's comments followed by our reply.

**Comments by referee #1, Jacqueline Caplan-Auerbach**
Received: 24 April 2018
This review is my second of this paper, and I appreciate the response that the authors gave to my first review. Many of my original questions have been well addressed in this version of the paper. I still have some concerns as well, and these (as well as areas in which my questions have been satisfied) are discussed below.
Previous concerns:
1. I wondered if the frequencies were actually harmonic: In this draft the authors describe specific frequencies (2.5, 5, 7.5 Hz), but I still find this very difficult to see. The frequency bands shown in the spectrogram are very narrow and it's hard to see if this is truly the frequency of the signal, or if it results from the length of the Fourier transform. Either way, I don't see that there are clear signals at 5 or 7.5 Hz. It is also very difficult for me to see how the red/blue lines drawn on the spectrogram correspond to the actual signal…I don't see that the lines correlate with actual frequencies in the spectrogram.
**Reply:** We agree that in the previous version of Figure S7 it wasn't clear what spectral peaks had this frequency content. For this reason, we've added another time slice to this figure that shows the 2.5/5/7.5 Hz signal. The reviewer is correct to point out that all of the spectral peaks are not in perfect harmonic relationship. As we've labeled in this figure, the frequencies of the three highest peaks are at 2.25, 4.20, and 7.13 Hz. The frequency bin size is 0.1 Hz in these spectrograms, which suggests that these measured frequencies are within plus/minus one frequency bin of being harmonic. We've updated the text to say "approximately harmonic" rather than suggest that there is a perfectly harmonic relationship.

Finally, the authors model upgliding and downgliding, but they do not comment on the feature that sweeps from 12 Hz to 14, and back down again. I don't think that all signals necessarily need to be modeled, but this one stands out in all of the data but goes unnoticed or unexamined.
**Reply:** We feel that these features are just one of many "higher order" observations that we are unable to describe within a reasonably sized manuscript. One reason

that we choose not to focus on these particular features is that they aren't clearly observed on as many stations as the lower-frequency spectral bands. We've included a statement in the text that informs the reader of our choice to focus on the clearest and most robust observations.

2. My previous review asked the authors to clarify when the landslide begins—this is now clear.
**Reply:** This clarification has been made already.

3. In my original review I expressed doubt that the slide could happen to have two asperities, both of which failed with sufficient regularity to generate identically repeating events, but one of which showed acceleration while the other showed deceleration. In this version the authors retain that model for one patch but assume the other changes by growing in size. While that is also plausible, it still feels like too great an assumption. I note that reviewer #2 in the first draft was concerned that assumptions are quickly taken to be truth, and I feel that this is a good description of this model—it still feels implausible.
**Reply:** This is a reasonable point. There is a fundamental uncertainty about whether our model is the correct one. Furthermore, even within the assumptions of our model possible explanations are not unique and there are tradeoffs between different parameters. For this reason, there is considerable uncertainty regarding the interpretation of the data. We have modified the text to express our view that there is both 1) uncertainty in model selection and 2) uncertainty and non-uniqueness within our choice of model. Despite these limitations, we do feel that our model is the only physics-based model currently available that is able to qualitatively match the observed gliding spectral peaks, even if there are quantitative limitations.

4. While the authors have done a good job making the force history more clear (I appreciate the figures showing these forces), I agree with the associate editor that it's concerning that the force history still has the landslide source at the top of the slide rather than the center of mass.
**Reply:** We appreciate this comment and agree that we cannot put the starting point at the top of the scarp and that it has to be at the center of mass of the initial sliding block. We have now revised the manuscript and the related Figure 5.

5. The authors added some important references, for which I thank them.
**Reply:** This modification has been made already.

6. In their response to the associate editor's comments, the authors state that the supplementary Figure 7, "shows that individual events are in fact discernable, but that they are rather jumbled in appearance and difficult to interpret". This puzzles me, because I don't see any discrete events within the seismogram. I'm not sure what they are referring to as individual events, but if they feel such events are existent, they should at least be pointed out.

**Reply**: We've modified supplementary Figure 7 to show the velocity pulses that we interpret as individual events.

I also agree with one of the other commenters (either reviewer #2 or the associate editor) that it doesn't make sense to have discrete events that are too small to be seen individually, yet that merge together to be recorded across the network (an argument made in the revision). It is far more compelling to consider that there never were discrete events before they were recorded as tremor.

**Reply:** We agree. The individual events are individually visible. Our claim is that they are more easily understood in the frequency domain than in the time domain. But the time and frequency domains are dual to each other. We have changed the manuscript to reflect this point.

I do think it's reasonable to interpret tremor as being overlapping events—that's certainly a commonly invoked mechanism. But I'm not sure there is sufficient evidence here to suggest that this is the best explanation for this particular tremor.

**Reply:** We agree (as discussed above) that there are considerable uncertainties associated with our model and interpretation. To our knowledge, however, there is no other physics-based model that explains the type of qualitative behavior observed at Askja, i.e. gliding spectral lines. Until some other model is proposed that is able to match the type of qualitative agreement that we have achieved, we feel that it would be unreasonable not to publish this possible explanation of the tremor process, albeit with adequate statements of the uncertainty and non-uniqueness associated with this interpretation.

To this point, we have attempted to thoroughly examine other possible tremor mechanisms. In Section 6.2, we have compiled what is to our knowledge a complete list of proposed tremor source processes. This list includes:

(i)   "fluid-flow-induced oscillations of conduit or fracture walls (Julian, 1994; Hellweg, 2000; Rust et al., 2008; Matoza et al., 2010; Corona-Romero et al., 2012; Dunham and Ogden, 2012; Unglert and Jellinek 2015);

(ii)  resonance of fluid-filled cracks or pipes with open or closed ends (Chouet, 1985, 1986, 1988; Benoit and McNutt, 1997; Jousset et al., 2003; Neuberg, 2006; Jellinek and Bercovici, 2011; Röösli et al., 2014; Sturton and Walter et al., 2015; Lipovsky and Dunham, 2015);

(iii) bubble growth or collapse due to hydrothermal boiling of groundwater (Leet, 1988; Kedar et al., 1998; Cannata et al., 2010); and

(iv)  continuously repeating processes such as stick-slip motion (Neuberg, 2000; Powell and Neuberg, 2003; Dmitrieva et al., 2013; Hotovec et al., 2013; Lipovsky and Dunham, 2016; see also reviews by McNutt, 1992 and Konstantinou and Schlindwein, 2003)."

In Section 6.2, we then move through these possible explanations and explain why we find repeating stick-slip to be the most plausible explanation.

In summary, some things were well corrected, and I very much appreciate the care that the authors took in responding to reviewer concerns. However, I'm still

unconvinced that the tremor signal comprises discrete events. I do find the signal fascinating and worthy of commentary, however.

**Reply:** We feel that the point about discrete events deserves clarification. It is not possible to have spectral peaks without repeating events and vice versa; that much is simply a fact about the Fourier transform. The relevant derivation related to the Fourier transform goes by several names including the "Dirac Comb". It is presented in numerous signal processing textbooks (e.g., Brigham) or, for example, on this Wikipedia page: https://en.wikipedia.org/wiki/Dirac_comb. As shown in Figure 7, spectral peaks are clearly present in the data. As discussed in a previous response, although the spectral peaks are not perfectly harmonic, they are in fact approximately harmonic. It is therefore a consequence of the definition of the Fourier transform that the corresponding time domain velocity seismogram will have an approximately periodically repeating pulse-like signal. The approximate nature of the harmonic relationship will result in small deviations from a perfect pulse shape and perfect repetition.

**Comments by referee #2, anonymous**
Received: 24 April 2018
Thank you very much for the modification. I have two major comments for your revised manuscript.
Section 3.2.2 I think it would be easier to understand if you can write in the following manner (which is how you computed): 1) present inverted force(Assuming a block model with a constant landslide mass over time, inverted force can be expressed as the product of the mass and time-series acceleration.) 2) explain how to obtain mass mass (Then, we find the most likely mass by...) 3) show the most probable mass value 4) velocity and displacement obtained from the force and estimated mass.
**Reply:** We have rewritten and rephrased parts of the waveform inversion to clarify the computing procedure.

P14 L18-22 "Although individual events are roughly discernible in seismograms, the effects of attenuation and superposition of multiple sources makes time domain analysis difficult (Fig. S7). Following previous studies, we therefore prefer spectral analysis 20 over the time domain (Dmitrieva et al., 2013; Hotovec at al., 2013; Winberry et al., 2013; Lipovsky and Dunham, 2016). We note that the tremor observed by Yamada et al. (2016) was observed at a much shorter source-to-station distance <1 km, whereas our closest station is 3.5 km from the landslide source area."
I think detectability of discrete events and the transmission of energy in the long-distance are different.
**Reply**: Yes, this is precisely the point we wanted to make. It is exactly our claim that the observed seismograms are the result of events that are blurred together yet still represent seismic energy that has been transmitted from the vicinity of the landslide to the receiver.

For example, the seismic station in Yamada et al. (2016) was <1km, but the stick-slip event was not recorded at any other stations (R~20km), whereas the signal was detectable even R~30km for the Askja event. This implies that the stronger attenuation or smaller amplitude in the example of Yamada et al. (2016), compared to this article. In the above sentences, it seems that the reason why the individual event was not detectable was due to the source-to-station distance. However, I think it is not true, since Poli (2017) showed clear individual events for R=30km, but your data did not even R~3.5km.

**Reply:** This is a reasonable point: Although distance factors into amplitude decay due to geometric spreading, distance is not the only factor that alters seismic amplitudes. It is also possible that the events at Askja were simply smaller or less energetic, two ideas that we quantify more precisely in the following. The seismic particle velocity amplitude of a double couple source is equal to $\ddot{M}/(4\pi\,\rho\,c^3 R)$, where R is the epicentral distance, $\rho$ is density, c is wave speed, $M = \pi\,r^2 D\,G$ is the seismic moment with slip D, shear modulus G, and fault radius r, and the over-dots denote time derivatives. With constant material properties, amplitudes are therefore expected to decay either as a result of increasing epicentral distance, or as a result of decreasing moment acceleration. Moment acceleration in turn, may decrease either because of reduced slip or fault radius, or because of reduced second derivatives of these quantities. The former effect, i.e., simply having a smaller absolute moment, provides a precise definition of what it means for an event to be smaller. The two latter effects result in reduced moment acceleration; these effects quantify what it means for an event to be less energetic. We now return to the comparison between the events at Askja and those studied by Poli (2017) and Yamada et al. (2016). In particular, we conclude that there are two plausible explanations for the appearance of clearly discernable individual events by Poli (2017) and Yamada et al. (2016) while such events are not "clearly discernable" at Askja: 1) reduced moment, 2) reduced event moment acceleration.

**Comments by K. Allstadt (Editor)**
Received: 24 April 2018
Both of the original reviewers have reviewed your revisions and are mostly satisfied with the changes you have made, but both still request minor revisions. There are three main areas that need to be addressed:
1) The model of a repeating source, while plausible, is presented with too much certainty given the evidence. The authors can and should still include this as a possible model but should also allow for other possibilities.

**Reply:** This is a fair point and we have responded in the text by elaborating in greater detail on the sources of uncertainty and non-uniqueness associated with out interpretation and modelling efforts.

2) Related, to point 1, the explanation given for the lack of observable discrete events is still not satisfactory to either reviewer. See their comments for details.

**Reply:** We have responded to this point in the above comments. We generally feel that interacting with this constructive criticism has improved the strength of the manuscript.

3) While improved, there are still some issues of clarity regarding the force history analysis, see suggestions in reviewer report #1. Additionally, the trajectory should track the landslide's center of mass location - while the current figure shows the trajectory ending where the authors think the center of mass of the deposits lies, it does not start at the center of mass of the source area where it should be, but rather at the uppermost edge of the slide. This needs to be fixed and all of the values derived from that trajectory (mass, velocities etc.) need to be updated.

**Reply:** We have changed the starting point of the centre of the block mass in the revised version of Figure 5 and made related changes in the manuscript.

[revised manuscript text omitted]

---

## Author Response (AR3)

**Authors' reply to comments of the editor**

Original title:                 Dynamics of the Askja caldera July 2014 landslide, Iceland, from seismic signal analysis: precursor, motion and aftermath

First submission:          30 November 2017
Manuscript No:            esurf-2017-68
First reply to reviews:   16 February 2018 submitted by A. Schöpa on behalf of the authors

We thank the editor for her constructive comments and suggestions. Please find in the following the editor's comments followed by our reply**.**

**Comments by K. Allstadt (Editor)**
Received: 24 April 2018
-On page 8, the authors use the word "arrest" improperly, the word that should be used for a landslide coming to a stop is "rest" -e.g., "...before coming to rest."
**Reply:** We changed the word as suggested.

-Also on page 8, the authors make a contradictory statement that the material going into the lake was much smaller than the total mass as justification for not considering the trajectory as continuing into the water as far as it was observed, but then in the next paragraph give a reported number for the volume of the submerged mass that was as high as a third of the total mass - that is a substantial portion of the mass. This contradiction needs to be addressed.
**Reply:** We are grateful for spotting this point and changed the wording of the text.

-Also on page 8, the authors describe the process for choosing the correct mass that gives the most-likely trajectory, but do not give a description of the criteria that was used to determine which trajectory of they many they computed was "best".
**Reply:** We clarified this point by mentioning: "the final trajectory is determined by minimising the misfit between the observed (1200 m used in this study) and computed run-out distances."

-on page 9, line 19, "as it is the case" should be changed to "as is the case" and on line 20, "which that" should be changed to "which"
**Reply:** We changed the words as suggested.

-page 12, line 19 - do the authors mean "high frequency" here?
**Reply:** Thank you for spotting this mistake. We changed the wording to "high frequency".

And also, please give a reason why a force history inversion cannot be done for high frequency signals.
**Reply:** As we mention in section 3.2, "long-period seismic waves radiated by landslides result from the cycle of unloading and reloading of the solid Earth (Fukao,

1995, Takei and Kumazawa, 1994). This broad loading cycle is produced by the bulk acceleration and deceleration of the landslide mass (Okal, 1990)." In section 6.1, we say that "the high-frequency signals are caused by the momentum exchanges of block impacts, and frictional processes within the moving slide and along its boundaries, especially when the moving mass traverses small-scale topographic features on the sliding base (cf. Dammeier et al., 2011; Allstadt, 2013)." We made this difference clear in the section about the seismic signals of the afterslides.

-Section 6.2 - this section is a great addition to the paper, but the authors discuss just tremor in general. Please be clear here and throughout that gliding tremor/gliding spectral lines are also commonly observed in volcanic settings and in relation to some of the models discussed.
**Reply:** We clarified this point in the manuscript, e.g. by stating: "although our focus is on the Askja landslide, it is worth considering whether volcanic activity could have been responsible for the precursory seismic tremor."

And also, on pg 15 line 22, consider whether the change in recurrence interval of discrete events is the cause of gliding spectral lines in all possible tremor models, not just those related to stick-slip.
**Reply:** We now also mention, e.g., the reasons for gliding spectral lines in resonance phenomena.

-page 15 line 10, change "montion have" to "motion has".
**Reply:** We changed the words as suggested.

-Section 6.3 - this discussion about a seismically-based early warning system for landslides, which requires identification of a pre-existing potential landslide source, will make readers question why then not use a warning system that provide a more direct observation of the landslide deformation, such as high-sample rate GPS on the slide mass. Be sure to stress what advantage the proposed approach might have.
**Reply:** We added advantages of seismically-based early warning systems to the manuscript.

-page 17 line 20 - why do you propose slow deformation "should produce a seismic signal" - don't the conditions have to be just right for stick-slip sliding to occur, as discussed in great detail in earlier sections of the paper? There are many landslides for which repeating precursory events were not observed despite relatively close seismic stations.
**Reply:** We clarified this point in the manuscript.

[revised manuscript text omitted]

**5 Afterslides**

During 8 hours after the main landslide, several other high-amplitude short-period signals of much lower amplitude were recorded (for example, at 23:41:05 UTC, Fig. 4d). Their waveforms are spindle-shaped with dominant frequencies of about 1–2 Hz. The signals are only visible at frequencies >1 Hz and a force history inversion is thus not possible as this method requires the record of long-period seismic signals resulting from the cycling unloading and reloading of the solid Earth by a moving mass (Fukao, 1995, Takei and Kumazawa, 1994). The high-amplitude short-period events lasted between a few seconds and a minute and have characteristics such as emergent onsets, slowly decaying tails, and triangularly shaped spectrograms that are indicative for slope failures (Norris, 1994; Dammeier et al., 2011; Burtin et al., 2013; Chen et al., 2013). We attribute these signals to smaller slope failures that occurred after the main landslide. This interpretation is in line with the concept that high-frequency signals of mass movements mainly result from block impacts and frictional processes within a slide or flow (cf. Dammeier et al., 2011; Allstadt, 2013). 
[revised manuscript text omitted]

Anne Schöpa 14/3/18 09:45

Anne Schöpa 2/3/18 14:24

Anne Schöpa 14/3/18 09:45

Anne Schöpa 2/3/18 14:25

Anne Schöpa 14/3/18 09:45

Anne Schöpa 2/3/18 14:27

Anne Schöpa 15/2/18 14:03

Anne Schöpa 2/3/18 14:27

Anne Schöpa 15/2/18 14:03

Anne Schöpa 21/2/18 14:57

Anne Schöpa 11/1/18 18:20

Anne Schöpa 15/2/18 14:04

Anne Schöpa 7/2/18 15:19

Anne Schöpa 16/2/18 18:03

Although our focus is on the Askja landslide, it is worth considering whether volcanic activity could have been responsible for the precursory seismic tremor. Mechanical analyses of hydraulic sources for seismic tremor showed that fluid-flow instabilities producing wall oscillations (Julian, 1994) require flow speeds on the order of the speed of sound (Dunham and Ogden, 2012), thus suggesting that the applicability of these physics is limited to situations such as high velocity volcanic jets. As the Askja landslide was not associated with any volcanic activity that would support this mechanical model of tremor generation through hydraulic processes, we conclude that a hydraulic source is unlikely to explain the phenomena observed at Askja.

Furthermore, Lipovsky and Dunham (2015) analysed seismic tremor due to hydraulic resonance and found that the resonant frequencies of a hydraulic fracture are expected to be unevenly spaced following $f_n/f_1 = n^{3/2}$. Complementary, Lipovsky and Dunham (2016) showed that a simple application of the Fourier transform to a repeating sequence of slip pulses results in a frequency pattern of $f_n/f_1 = n$. For a fundamental tone of $f_1 = 2.3$ Hz as observed for the Askja landslide tremor, we would expect its first harmonic at 6.5 Hz for a resonating fracture or at 4.6 Hz for a stick-slip source. Observations of the Askja landslide tremor show that the spectral peaks are relatively evenly spaced with $f_1 = 2.3$ Hz, $f_2 = 4.3$ Hz, and $f_3 = 7.1$ Hz, a pattern that is in closer agreement with the harmonic relationship $f_n/f_1 = n$. This provides observational evidence for a stick-slip mechanism and against a hydraulic source of the tremor before the Askja landslide.

Several additional lines of reasoning support the interpretation of the Askja landslide tremor as being due to repeating stick-slip motion along the landslide failure plane. Stick-slip motion has been observed as precursors to other landslides (Yamada et al., 2016; Poli, 2017), although in these cases individual stick-slip events could easily be distinguished. Although individual events are roughly discernible in seismograms, the effects of attenuation and superposition of multiple sources makes time domain analysis difficult (Fig. S7). Following previous studies, we therefore prefer spectral analysis over the time domain (Dmitrieva et al., 2013; Hotovec at al., 2013; Winberry et al., 2013; Lipovsky and Dunham, 2016). We note that the tremor observed by Yamada et al. (2016) was observed at a much shorter source-to-station distance <1 km, whereas our closest station is 3.5 km from the landslide source area. It is also possible that individual stick-slip events were more clearly visible in the studies by Yamada et al. (2016) and Poli (2017) because the events were either larger or were more energetic. We note that stick-slip motion has previously been proposed to cause seismic tremor on the sliding planes of sudden surface mass movements including ice-rock avalanches (Caplan-Auerbach et al., 2004, Huggel et al., 2008) and during glacier sliding (Caplan-Auerbach and Huggel, 2007; Winberry et al., 2013; Allstadt and Malone, 2014; Helmstetter et al., 2015; Lipovsky and Dunham, 2016).

When tremor occurs due to repeating stick-slip cycles, gliding of the frequency bands is the result of a changing recurrence time (Lockner et al., 1991; Neuberg 2000; Dmitrieva et al., 2013; Hotovec et al., 2013). This is in contrast to the occurrence of frequency bands due to resonance phenomena where changing frequency contents are mainly caused by variations of the resonator's geometry and the fluid's properties (e.g., 
[revised manuscript text omitted]
 such as high-rate GPS located directly on the landslide. In addition, seismically-based observations can help identifying the landslide's source mechanisms and properties, and the failure sequence including precursory activity and aftermath, which yields a comprehensive concept of the event by using only one system. While seismic landslide early-warning systems may not be possible at the present time, our goal here is simply to outline several scientific and engineering considerations that must be addressed to better understand the feasibility of such a system.

First, future observations should be made to determine whether accelerating stick-slip, manifested as either isolated events (e.g., Yamada et al., 2016; Poli, 2017) or as seismic tremor (e.g., as before the Askja landslide), is in fact a sufficiently common precursor to large scale slope failures. There is evidence that this may be the case. Many voluminous slope failures do start as slow-moving landslides (Palmer, 2017). The theory presented in Section 4 predicts that at low sliding rates, repeating stick-slip events will have longer inter-event times. Future work could attempt to establish bounds for the observability of small, infrequently repeating events that might be near the noise level. Furthermore, some already-monitored

Anne Schöpa 16/2/18 11:59

Anne Schöpa 7/2/18 15:04

Anne Schöpa 7/2/18 15:05

Anne Schöpa 7/2/18 15:05

[revised manuscript text omitted]